# Robust estimation of bacterial cell count from optical density

Jacob Beal [1✉], Natalie G. Farny [2✉], Traci Haddock-Angelli [3✉], Vinoo Selvarajah[3], Geoff S. Baldwin [4✉], Russell Buckley-Taylor[4], Markus Gershater[5✉], Daisuke Kiga[6], John Marken[7], Vishal Sanchania[5], Abigail Sison[3], Christopher T. Workman [8✉] & the iGEM Interlab Study Contributors*

Optical density (OD) is widely used to estimate the density of cells in liquid culture, but cannot be compared between instruments without a standardized calibration protocol and is challenging to relate to actual cell count. We address this with an interlaboratory study comparing three simple, low-cost, and highly accessible OD calibration protocols across 244 laboratories, applied to eight strains of constitutive GFP-expressing *E. coli*. Based on our results, we recommend calibrating OD to estimated cell count using serial dilution of silica microspheres, which produces highly precise calibration (95.5% of residuals <1.2-fold), is easily assessed for quality control, also assesses instrument effective linear range, and can be combined with fluorescence calibration to obtain units of Molecules of Equivalent Fluorescein (MEFL) per cell, allowing direct comparison and data fusion with flow cytometry measurements: in our study, fluorescence per cell measurements showed only a 1.07-fold mean difference between plate reader and flow cytometry data.

[1] Raytheon BBN Technologies, Cambridge, MA, USA. [2] Department of Biology and Biotechnology, Worcester Polytechnic Institute, Worcester, MA, USA. [3] iGEM Foundation, Cambridge, MA, USA. [4] Department of Life Sciences and IC-Centre for Synthetic Biology, Imperial College London, London, UK. [5] Synthace, London, UK. [6] Faculty of Science and Engineering, School of Advanced Science and Engineering, Waseda University, Tokyo, Japan. [7] Department of Bioengineering, California Institute of Technology, Pasadena, CA, USA. [8] DTU-Bioengineering, Technical University of Denmark, Kongens Lyngby, Denmark. *A list of authors and their affiliations appears at the end of the paper. ✉email: jakebeal@ieee.org; nfarny@wpi.edu; traci@igem.org; g.baldwin@imperial.ac.uk; m.gershater@synthace.com; cwor@dtu.dk

 **1**

Comparable measurements are a sine qua non for both science and engineering, and one of the most commonly needed measurements of microbes is the number (or concentration) of cells in a sample. The most common method for estimating the number of cells in a liquid suspension is the use of optical density measurements (OD) at a wavelength of 600 nm (OD600)[1]. The dominance of OD measurements is unsurprising, particularly in plate readers, as these measurements are extremely fast, inexpensive, simple, relatively non-disruptive, high-throughput, and readily automated. Alternative measurements of cell count—microscopy (with or without hemocytometer), flow cytometry, colony-forming units (CFU), and others, e.g., see refs. [2–5]—lack many of these properties, though some offer other benefits, such as distinguishing viability and being unaffected by cell states such as inclusion body formation, protein expression, or filamentous growth[6].

A key shortcoming of OD measurements is that they do not actually provide a direct measure of cell count. Indeed, OD is not even linearly related to cell count except within a limited range[7]. Furthermore, because the phenomenon is based on light scatter rather than absorbance, it is relative to the configuration of a particular instrument. Thus, in order to relate OD measurements to cell count—or even just to compare measurements between instruments and experiments—it is necessary to establish a calibration protocol, such as comparison to a reference material.

While the problems of interpreting OD values have been studied (e.g., refs. [1,6,7]), no previous study has attempted to establish a standard protocol to reliably calibrate estimation of cell count from OD. To assess reliability, it is desirable to involve a large diversity of instruments and laboratories, such as those participating in the International Genetically Engineered Machines (iGEM) competition[8], where hundreds of teams at the high school, undergraduate, and graduate levels have been organized previously to study reproducibility and calibration for fluorescence measurements in engineered E. coli[9,10]. As iGEM teams have a high variability in training and available resources, organizing an interlaboratory study with iGEM also demands that protocols be simple, low cost, and highly accessible. The large scale and high variability between teams also allows investigation of protocol robustness, as well as how readily issues can be identified and debugged in protocol execution.

We thus organized a large-scale interlaboratory study within iGEM to compare three candidate OD calibration protocols: a colony-forming unit (CFU) assay, the de facto standard assay for determining viable cell count; comparison with colloidal silica (LUDOX) and water, previously used for normalizing fluorescence measurements[9]; and serial dilution of silica microspheres, a new protocol based on a recent study of microbial growth[7]. Overall, this study demonstrates that serial dilution of silica microspheres is by far the best of these three protocols under the conditions tested, allowing highly precise, accurate, and robust calibration that is easily assessed for quality control and can also evaluate the effective linear range of an instrument. We thus recommend use of silica microsphere calibration within the linear range of OD measurements for cells with compact shape and matching refractive index. Adoption of this recommendation is expected to enable effective use of OD data for estimation of cell count, comparison of plate reader measurements with single-cell measurements such as flow cytometry, improved replicability, and better cross-laboratory comparison of data.

## Results
To evaluate the three candidate OD calibration protocols, we organized an interlaboratory study as part of the 2018 International Genetically Engineered Machine (iGEM) competition.

The precision and robustness of each protocol is assessed based on the variability between replicates, between reference levels, and between laboratories. The overall efficacy of the protocols was then further evaluated based on the reproducibility of cross-laboratory measurements of cellular fluorescence, as normalized by calibrated OD measurements.

**Experimental data collection.** Each contributing team was provided with a set of calibration materials and a collection of eight engineered genetic constructs for constitutive expression of GFP at a variety of levels. Specifically, the constructs consisted of a negative control, a positive control, and six test constructs that were identical except for promoters from the Anderson library[11], selected to give a range of GFP expression (illustrated in Fig. 1a, with complete details provided in Supplementary Data 1). In particular, the positive and negative controls and the J23101, J23106, and J23117 promoters were chosen based on their prior successful use in the 2016 iGEM interlaboratory study[9] as controls and "high", "medium", and "low" test levels, respectively. Beyond these, J23100 and J23104 were chosen as potential alternatives for J23101 (about which there were previous reports of difficulty in transformation), and J23116 was chosen as an intermediate value in the large gap in expression levels between J23106 and J23117 (expected values were not communicated to teams, however). These materials were then used to follow a calibration and cell measurement protocol (see the "Methods" section; Supplementary Note: Plate Reader and CFU Protocol and Supplementary Note: Flow Cytometer Protocol).

Each team transformed E. coli K-12 DH5-alpha with the provided genetic constructs, culuring two biological replicates for each of the eight constructs. Teams measured absorbance at 600 nm (OD600) and GFP in a plate reader from four technical replicates per biological replicate (for a total of eight replicates and fitting on a 96-well plate) at the 0 and 6 h time points, along with media blanks, thus producing a total of 144 OD600 and 144 GFP measurements per team. Six hours was chosen as a period sufficient for exponential growth, and the zero-hour measurement used only for comparison to exclude samples that failed to grow well. Teams with access to a flow cytometer were asked to also collect GFP and scatter measurements for each sample, plus a sample of SpheroTech Rainbow Calibration Beads[12] for fluorescence calibration.

Measurements of GFP fluorescence were calibrated using serial dilution of fluorescein with PBS in quadruplicate, using the protocol from ref. [9], as illustrated in Fig. 1b. Starting with a known concentration of fluorescein in PBS means that there is a known number of fluorescein molecules per well. The number of molecules per arbitrary fluorescence unit can then be estimated by dividing the expected number of molecules in each well by the measured fluorescence for the well; a similar computation can be made for concentration.

Measurements of OD via absorbance at 600 nm (OD600) were calibrated using three protocols and for each of these a model was devised for the purpose of fitting the data obtained in the study (Methods):

Calibration to colony-forming units (CFU), illustrated in Fig. 1c: Four overnight cultures (two each of positive and negative controls), were sampled in triplicate, each sample diluted to 0.1 OD, then serially diluted, and the final three dilutions spread onto bacterial culture plates for incubation and colony counting (a total of 36 plates per team). The number of CFU per OD per mL is estimated by multiplying colony count by dilution multiple. This protocol has the advantage of being well established and insensitive to non-viable cells and debris, but the disadvantages of an unclear number of cells per CFU, potentially high statistical

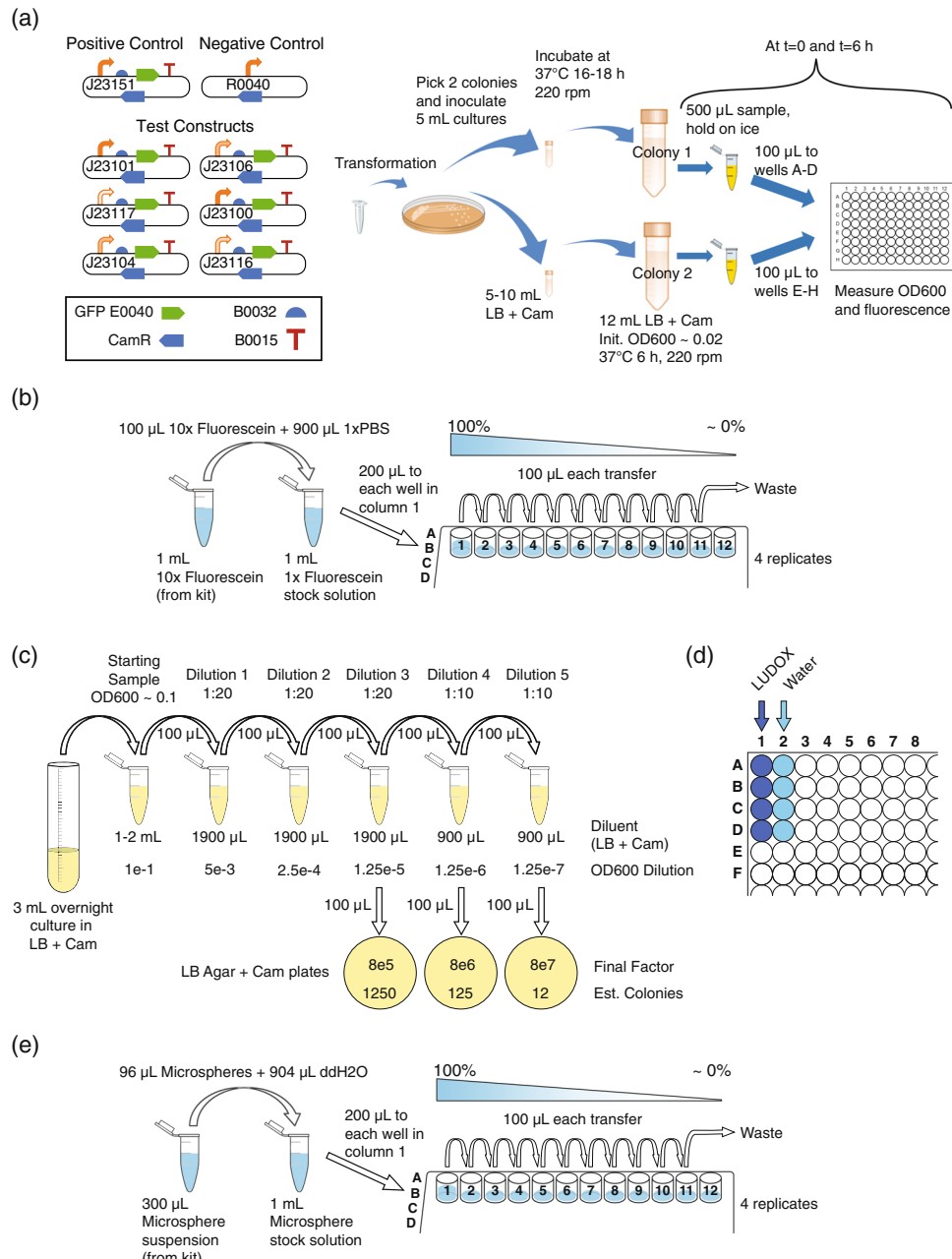

**Fig. 1 Study design. a** Each team cultured eight strains of engineered *E. coli* expressing GFP at various levels: positive and negative controls plus a library of six test constructs with promoters selected to give a range of levels of expression. Each team also collected four sets of calibration measurements, **b** fluorescein titration for calibration of GFP fluorescence, plus three alternative protocols for calibration of absorbance at 600 nm: **c** dilution and growth for colony-forming units (CFU), **d** LUDOX and water, and **e** serial dilution of 0.961 μm-diameter monodisperse silica microspheres.

variability when the number of colonies is low, and being labor intensive.

Comparison of colloidal silica (LUDOX CL-X) and water, illustrated in Fig. 1d: This protocol is adapted from ref. [9] by substitution of a colloidal silica formulation that is more dense and freeze-tolerant (for easier shipping). Quadruplicate measurements are made for both LUDOX CL-X and water, with conversion from arbitrary units to OD measurement in a standard spectrophotometer cuvette estimated as the ratio of their difference to the OD measurement for LUDOX CL-X in a reference spectrophotometer. This protocol has the advantage of using extremely cheap and stable materials, but the disadvantage that LUDOX CL-X provides only a single reference value, and that it calibrates for instrument differences in determination of

OD but cannot be used to estimate the number of cells, as all grades of LUDOX particles are far smaller than cells (<50 nm).

Comparison with serial dilution of silica microspheres, illustrated in Fig. 1e: This new protocol, inspired by the relationship between particle size, count, and OD[7], uses quadruplicate serial dilution protocol of 0.961-μm-diameter monodisperse silica microspheres in water, similar to fluorescein dilution, but with different materials. These particles are selected to match the approximate volume and optical properties of *E. coli*, with the particles having a refractive index of 1.4 (per manufacturer specification) and typical *E. coli* ranging from 1.33 to 1.41[7]. With a known starting concentration of particles, the number of particles per OD600 unit is estimated by dividing the expected number of particles in each well by the measured OD for

the well. This protocol has the advantages of low cost and of directly mapping between particles and OD, but the disadvantage that the microspheres tend to settle and are freeze-sensitive.

Data from each team were accepted only if they met a set of minimal data quality criteria (Supplementary Note: Data Acceptance Criteria), including values being non-negative, the positive control being notably brighter than the negative control, and measured values for calibrants decreasing as dilution increases. In total, 244 teams provided data meeting these minimal criteria, with 17 teams also providing usable flow cytometry data. Complete anonymized data sets and analysis results are available in Supplementary Data 2.

**Robustness of calibration protocols**. We assessed the robustness of the calibration protocols under test in two ways: replicate precision and residuals. Replicate precision can be evaluated simply in terms of the similarity of values for each technical replicate of a protocol. The smaller the coefficient of variation (i.e., ratio of standard deviation to mean), the more precise the protocol. With regards to residuals, on the other hand, we considered the modeled mechanism that underlies each calibration method and assess how well it fits the data. Here, the residual is the distance between each measured value provided by a team and the predicted value of a model fit using that same set of data (see Methods for details of each mechanism model and residual calculations). The smaller the residual value, the more precise the protocol. Moreover, the more similar the replicate precision and residuals across teams, the more robust the protocol is to variations in execution conditions.

Figure 2 shows the distribution of the coefficients of variation (CVs) for all valid replicates for each of the calibrant materials (see Methods for validity criteria). For CFU, basic sampling theory implies that the dilution with the largest number of countably distinct colonies (lowest dilution) should have the best CV, and indeed this is the case for 81.6% of the samples. This percentage is surprisingly low, however, and indicates a higher degree of variation than can be explained by the inherent stochasticity of the protocol: CFU sampling should follow a binomial distribution and have a little over 3-fold higher CV with each 10-fold dilution, but on average it was much less. This indicates the presence of a large component of variation with an

unknown source, which is further confirmed by the fact that even the best CVs are quite high: the best of the three dilutions for each team has CV ≤ 0.1 for only 2.1% of all data sets and CV ≤ 0.2 for only 16.4% of all data sets.

LUDOX and water have the lowest CV, at CV ≤ 0.1 for 86.9% (LUDOX) and 88.1% (water) of all replicate sets and CV ≤ 0.2 for 97.1% (LUDOX) and 98.0% (water) of all replicate sets. Microspheres and fluorescein have slightly higher CV, at CV ≤ 0.1 for 80.8% (microspheres) and 76.9% (fluorescein) of all replicate sets and CV ≤ 0.2 for 93.9% (microspheres) and 92.4% (fluorescein) of all replicate sets. The difference between these two pairs likely derives from the fact that the LUDOX and water samples are each produced in only a single step, while the serial dilution of microspheres and fluorescein allows inaccuracies to compound in the production of later samples.

The accuracy of a calibration protocol is ultimately determined by how replicate data sets across the study are jointly interpreted to parameterize a model of the calibration protocol, one part of which is the scaling function that maps between arbitrary units and calibrated units. As noted above, this can be assessed by considering the residuals in the fit between observed values and their fit to the protocol model. To do this, we first estimated the calibration parameters from the observed experimental values (see Methods for the unit scaling computation for each calibration method), then used the resulting model to "predict" what those values should have been (e.g., 10-fold less colonies after a 10-fold dilution). The closer the ratio was to one, the more the protocol was operating in conformance with the theory supporting its use for calibration, and thus the more likely that the calibration process produced an accurate value.

Here we see a critical weakness of the LUDOX/water protocol: the LUDOX and water samples provide only two measurements, from which two model parameters are set: the background to subtract (set by water) and the scaling between background-subtracted LUDOX and the reference OD. Thus, the dimensionality of the model precisely matches the dimensionality of the experimental samples, and there are no residuals to assess. As such, the LUDOX/water protocol may indeed be accurate, but its accuracy cannot be empirically assessed from the data it produces. If anything goes wrong in the reagents, protocol execution, or instrument, such problems cannot be detected unless they are so great as to render the data clearly invalid (e.g., the OD of water being less than the OD of LUDOX).

The CFU protocol and the two serial dilution protocols, however, both have multiple dilution levels, overconstraining the model and allowing likely accuracy to be assessed. Figure 3 shows the distribution of residuals for these three protocols, in the form of a ratio between the observed mean for each replicate set and the value predicted by the model fit across all replicate sets. The CFU protocol again performs extremely poorly, as we might expect based on the poor CV of even the best replicates: only 7.3% of valid replicate sets have a residual within 1.1-fold, only 14.0% within 1.2-fold, and overall the geometric standard deviation of the residuals is 3.06-fold—meaning that values are only reliable to within approximately two orders of magnitude! Furthermore, the distribution is asymmetric, suggesting that the CFU protocol may be systematically underestimating the number of cells in the original sample. The accuracy of the CFU protocol thus appears highly unreliable.

The microsphere dilution protocol, on the other hand, produced much more accurate results. Even with only a simple model of perfect dilution, the residuals are quite low (red line in Fig. 3b), having 61.0% of valid replicates within 1.1-fold, 83.6% within 1.2-fold, and an overall geometric standard deviation of 1.152-fold. As noted above, however, with serial dilution we may expect error to compound systematically with each dilution, and

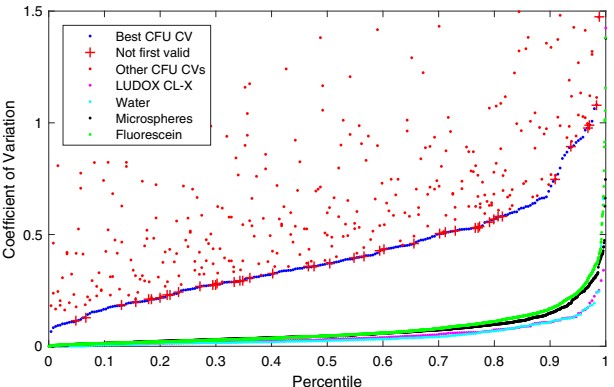

**Fig. 2 Distribution of the coefficient of variation for valid replicate sets in CFU, LUDOX/water, microspheres, and fluorescein (all teams included).** CFU models are generated from only the best CV dilution (blue); other dilutions are shown separately above. Even the best CV CFU dilutions, however, have a distribution far worse than the other four methods, and are surprisingly often not the lowest dilution (red crosses). Of the others, LUDOX (magenta) and water (light blue) have the best and near-identical distributions, while microspheres (black) and fluorescein (green) are only slightly higher.

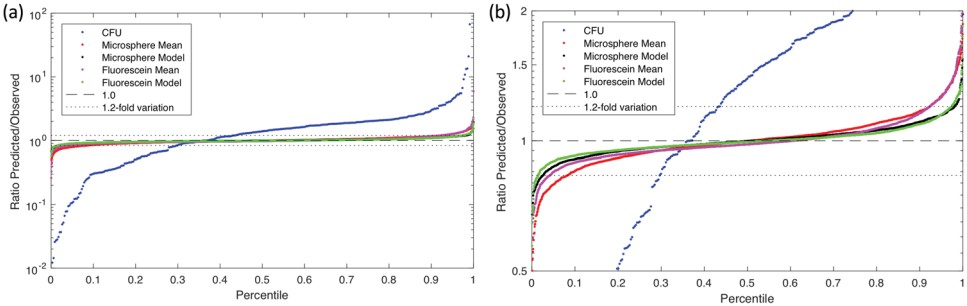

**Fig. 3 Distribution of residuals. a** Model fit residual distribution for each replica set in the CFU (blue), microsphere, and fluorescein calibration protocols (all teams included). **b** Expanding the Y axis to focus on the microsphere and fluorescein distributions shows that incorporating a model parameter for systematic pipetting error (black, green) produces a notably better fit (and thus likely more accurate unit calibration) than a simple geometric mean over scaling factors (red, magenta).

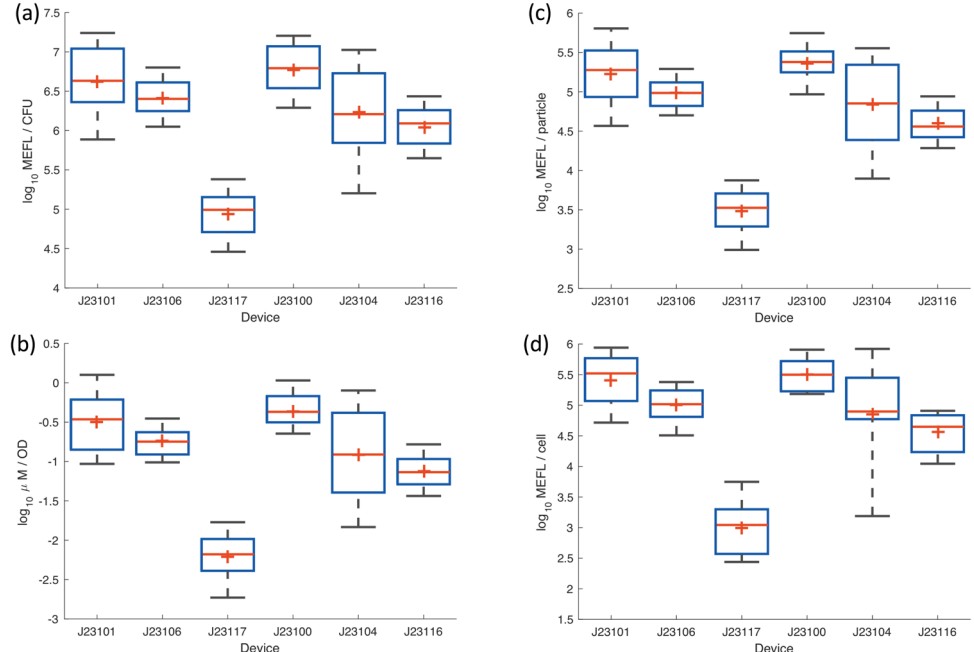

**Fig. 4 Measured fluorescence of test devices.** Measured fluorescence of test devices after 6 h of growth using **a** CFU calibration, **b** LUDOX/water calibration, **c** microsphere dilution calibration, and **d** flow cytometry. In each box, red plus indicates geometric mean, red line indicates median, top and bottom edges indicate 25th and 75th percentiles, and whiskers extend from 9 to 91%. Team count per condition provided in Supplementary Data 3.

indeed the value sequences in individual data sets do tend to show curves indicative of systematic pipetting error. When the model is extended to include systematic pipetting error (see Methods subsection on "Systematic pipetting error model"), the results improve markedly (black line in Fig. 3b), to 82.4% of valid replicates within 1.1-fold, 95.5% within 1.2-fold, and an overall geometric standard deviation improved to 1.090-fold. Fluorescein dilution provides nearly identical results: with a perfect dilution model (magenta line in Fig. 3b), having 71.1% of valid replicates within 1.1-fold, 88.2% within 1.2-fold, and an overall geometric standard deviation of 1.148-fold, and systematic pipetting error improving the model (green line in Fig. 3b), to 88.1% of valid replicates within 1.1-fold, 98.0% within 1.2-fold, and an overall geometric standard deviation of 1.085-fold.

Based on an analysis of the statistical properties of calibration data, we may thus conclude that the microsphere and fluorescein dilution protocols are highly robust, producing results that are precise, likely to be accurate, and readily assessed for execution quality on the basis of calibration model residuals. The LUDOX/water protocol is also highly precise and may be accurate, but its execution quality cannot be directly assessed due to its lack of

residuals. The CFU protocol, on the other hand, appears likely to be highly problematic, producing unreliable and likely inaccurate calibrations.

**Reproducibility and accuracy of cell-count estimates**. Reproducibility and accuracy of the calibration protocols can be evaluated through their application to calibration of fluorescence from *E. coli*, as normalized by calibrated OD measurements. Figure 4 shows the fluorescence values computed for each of the three fluorescence/OD calibration combinations, as well as for calibrated flow cytometry, excluding data with poor calibration or outlier values for colony growth or positive control fluorescence (for details see Methods on determining validity of *E. coli* data). Overall, the lab-to-lab variation was workably small, with the geometric mean of the geometric standard deviations for each test device being 2.4-fold for CFU calibration, 2.21-fold for LUDOX/water calibration, and 2.21-fold for microsphere dilution calibration. These values are quite similar to those previously reported in ref. [9], which reported a 2.1-fold geometric standard deviation for LUDOX/water.

Note that these standard deviations are also dominated by the high variability observed in the constructs with J23101 and J23104, both of which appear to have suffered notable difficulties in culturing, with many teams' samples failing to grow for these constructs, while other constructs grew much more reliably (see Supplementary Fig. 1). Omitting the problematic constructs finds variations of 2.02-fold for CFU calibration, 1.84-fold for LUDOX/ water calibration, and 1.83-fold for microsphere dilution calibration. Flow cytometry in this case is also similar, though somewhat higher variability in this case, at 2.31-fold (possibly due to the much smaller number of replicates and additional opportunities for variation in protocol execution). All together, these values indicate that, when filtered using quality control based on the replicate precision and residual statistics established above, all three OD calibration methods are capable of producing highly reproducible measurements across laboratories.

To determine the accuracy of cell-count estimates, we compared normalized bulk measurements (total fluorescence divided by estimated cell count) against single-cell measurements of fluorescence from calibrated flow cytometry, which provides direct measurement of per-cell fluorescence without the need to estimate cell count (see Methods on "Flow cytometry data processing" for analytical details). In this comparison, an accurate cell count is expected to allow bulk fluorescence measurement normalized by cell count to closely match the per-cell fluorescence value produced by flow cytometry. In making this comparison, there are some differences that must be considered between the two modalities. Gene expression typically has a log-normal distribution[13], meaning that bulk measurements will be distorted upward compared to the geometric mean of log-normal distribution observed with the single-cell measurements of a flow cytometer. In this experiment, for typical levels of cell-to-cell variation observed in *E. coli*, this effect should cause the estimate of per-cell fluorescence to be approximately 1.3-fold higher from a plate reader than a flow cytometer. At the same time, non-cell particles in the culture will tend to distort fluorescence per-cell estimates in the opposite direction for bulk measurement, as these typically contribute to OD but not fluorescence in a plate reader, but the vast majority of debris particles are typically able to be gated out of flow cytometry data. With generally healthy cells in log-phase growth, however, the levels of debris in this experiment are expected to be relatively low. Thus, these two differences are likely to both be small and in opposite directions, such that we should still expect the per-cell fluorescence estimates of plate reader and flow cytometry data to closely match if accurately calibrated.

Of the three OD calibration methods, the LUDOX/water measurement is immediately disqualified as it calibrates only to a relative OD, and thus cannot produce comparable units. Comparison of CFU and microsphere dilution to flow cytometry is shown in Fig. 5. The CFU-calibrated measurements are far higher than the values produced by flow cytometry, a geometric mean of 28.4-fold higher, indicating that this calibration method badly underestimates the number of cells. It is unclear the degree to which this is due to known issues of CFU, such as cells adhering into clumps, as opposed to the problems with imprecision noted above or yet other possible unidentified causes. Whatever the cause, however, CFU calibration is clearly problematic for obtaining anything like an accurate estimate of cell count.

Microsphere dilution, on the other hand, produces values that are remarkably close to those for flow cytometry, a geometric mean of only 1.07-fold higher, indicating that this calibration method is quite accurate in estimating cell count. Moreover, we may note that the only large difference between values comes with the extremely low fluorescence of the J23117

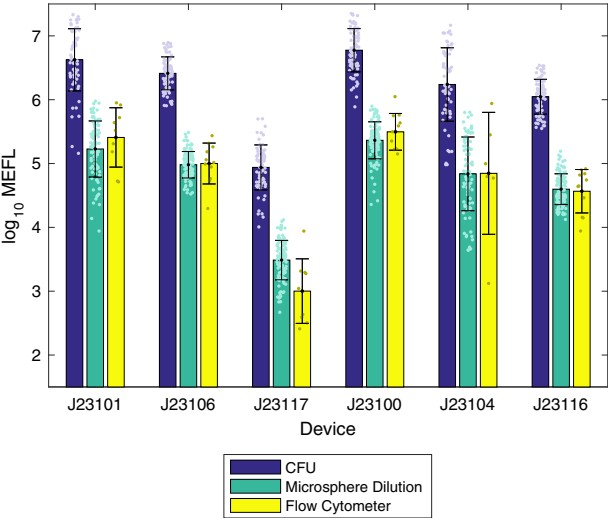

**Fig. 5 Fluorescence per cell after 6 h of growth, comparing calibrated flow cytometry to estimates using cell count from CFU and microsphere dilution protocols (LUDOX/water is not shown as the units it produces are not comparable).** Microsphere dilution produces values extremely close to the ground truth provided by calibrated flow cytometry, whereas the CFU protocol produces values more than an order of magnitude different, suggesting that CFU calibration greatly underestimates the number of cells in the sample. Bars show geometric mean and standard deviation. Team count per condition provided in Supplementary Data 3.

construct, which is unsurprising given that flow cytometers generally have a higher dynamic range than plate readers, allowing better sensitivity to low signals.

## Discussion

Reliably determining the number of cells in a liquid culture has remained a challenge in biology for decades. For the field of synthetic biology, which seeks to engineer based on standardized biological measurements, it was critical to find a solution to this challenge. Here, we have compared the most common method for calibrating OD to cell number (calculation of CFU) to two alternative methods of calibration: LUDOX/water and microsphere serial dilution. The qualitative and quantitative benefits and drawbacks of these three methods for OD calibration are summarized in Table 1.

These three protocols are all inexpensive, with the reagent cost for both LUDOX/water and microsphere serial dilution being < $0.10 US. The CFU protocol has well-known issues of cell clumping and slow, labor-intensive execution, and counts only live and active cells, which can be either a benefit or a limitation depending on circumstances, though it does benefit from being insensitive to cell shape and optical properties. In addition, the CFU counts in this study exhibited a remarkably high level of variability, which may call into question the use of the CFU method as a standard for determining cell counts. This observed variability is not without precedent—prior work has also demonstrated *E. coli* CFU counting performing poorly on measures of reproducibility and repeatability in an interlaboratory study[14].

The microsphere protocol, on the other hand, has no major drawbacks and provides a number of notable benefits when applied to cells with shapes and optical properties that can be reasonably approximated with appropriately sized microspheres. First, the microsphere protocol is highly robust and reliable, particularly compared with CFU assays. Second, failures are much easier to diagnose with the microsphere protocol, since it

**Table 1 Summary of the benefits and drawbacks of the three calibration protocols.**

| Protocol | Benefits | Drawbacks/limitations |
|---|---|---|
| Colony-forming units (CFU) | Inexpensive | Lower precision |
| | Requires no additional reagents | Count affected by cell clumping/adhesion |
| | Counts only live and active cells, eliminating quiescent cells, dead cells, and debris | |
| | Not affected by cell shape, optics | Labor intensive |
| | | Slow (overnight incubation) |
| LUDOX/water | Extremely simple, fast, and cheap | Generates only a single calibration point |
| | High precision | Cell count is still relative |
| Microsphere serial dilution | Inexpensive | Slightly more difficult to perform, as it must be completed before spheres have time to settle |
| | Highest precision | |
| | Many dilution levels helps with quality control and corrections | |
| | Also assesses linear range of instrument | |

has many distinct levels that can be compared. This is particularly salient when compared with the LUDOX/water protocol, which only provides a single calibration point at low absorbance (and thus susceptible to instrument range issues), and to the CFU protocol, where failures may be difficult to distinguish from inherent high variability. With the microsphere protocol, on the other hand, some failures such as systematic dilution error and instrument saturation can not only be detected, but also modeled and corrected for. Finally, the microsphere protocol also permits a unit match between plate reader and flow cytometry measurements (both in cell number and in fluorescence per cell), which is highly desirable, allowing previously impossible data fusion between these two complementary platforms (e.g., to connect high-resolution time-series data from a plate reader with high-detail data about population structure from a flow cytometer). Accordingly, based on the results of this study, we recommend the adoption of silica microsphere calibration for robust estimation of bacterial cell count. As long as OD measurements are within the linear range, this calibration protocol is expected to enable effective use of OD data for estimation of actual cell count, comparison of plate reader measurements with single-cell measurements such as flow cytometry, improved replicability, and better cross-laboratory comparison of data.

With regards to future opportunities for extension, we note that these methods seem likely to be applicable to other instruments that measure absorbance (e.g., spectrophotometers, automated culture flasks) by appropriately scaling volumes and particle densities. Similarly, it should be possible to adapt to other cell types by selecting other microspheres with appropriately adjusted diameters and materials for their optical properties (indeed, per ref. [7], many other commonly used bacteria have quite similar refractive index values), and a wide range of potential options are already readily available from commercial suppliers. Finally, further investigation would be valuable for more precisely establishing the relationship between cell count and particle count. It would also be useful to quantify the degree to which the estimates are affected by factors such as changing optical properties associated with cell state, distribution, shape, and clustering, and to investigate means of detecting and compensating for such effects.

## Methods

Participating iGEM teams measured OD and fluorescence among the same set of plasmid-based devices, according to standardized protocols. In brief, teams were provided a test kit containing the necessary calibration reagents, a set of standardized protocols, and pre-formatted Excel data sheets for data reporting. Teams provided their own plate reader instruments, consumables/plasticware, competent

E. coli cells, PBS, water, and culture medium. First, teams were asked to complete a series of calibration measurements by measuring LUDOX and water, and also making a standard curve of both fluorescein and silica microspheres. Next, each team transformed the plasmid devices into E. coli and selected transformants on chloramphenicol plates. They selected two colonies from each plate to grow as liquid cultures overnight, then the following day diluted their cultures and measured both fluorescence and OD after 0 and 6 h of growth. Some of these cultures were also used to make serial dilutions for the CFU counting experiment. Teams were asked to report details of their instrumentation, E. coli strains used, and any variations from the protocol using an online survey. Additional details are available in the Supplementary Information.

**Calibration materials**. The following calibration materials were provided to each team as a standard kit: 1 ml of LUDOX CL-X (Sigma-Aldrich, #420891) and 1.00e−8 moles of fluorescein (Sigma-Aldrich, #46970). About 300 μl of 0.961-μm-diameter monodisperse silica beads (Cospheric, SiO$_2$MS-2.0, 0.961 μm) in ddH$_2$O were prepared to contain 3.00e8 beads.

Fluorescein samples tubes were prepared with 1.00e−8 moles fluorescein in solution in each tube, which was then vacuum dried for shipping. Resuspension in 1 ml PBS would thus produce a solution with initial concentration of 10 μM fluorescein.

Each team providing flow cytometry data also obtained their own sample of SpheroTech RCP-30-5A Rainbow Calibration Particles (SpheroTech). A sample of this material is a mixture of particles with eight levels of fluorescence, which should appear as up to eight peaks (typically some are lost to saturation on the instrument). Teams used various different lots, reporting the lot number to allow selection of the appropriate manufacturer-supplied quantification for each peak.

**Constructs, culturing, and measurement protocols**. The genetic constructs supplied to each team for transformation are provided in Supplementary Data 1. The protocol for plate readers, exactly as supplied to each participating team, is provided in Supplementary Note: Plate Reader and CFU Protocol. The supplementary protocol for flow cytometry is likewise provided in Supplementary Note: Flow Cytometer Protocol.

**Criteria for valid calibrant replicates**. For purpose of analyzing the precision of calibrants, the following criteria were used to determine which replicate sets are sufficiently valid for inclusion of analysis:

CFU: A dilution level is considered valid if at least 4 of the 12 replicate plates have a number of colonies that are >0 but not too numerous to count (participants were instructed they could report anything over 300 colonies to be too numerous to count). A calibration set is considered valid if there is at least one valid dilution level. Of the 244 data sets, 241 are valid and 3 are not valid.

LUDOX/water: A LUDOX/water calibration is considered valid if it fits the acceptance criteria in Supplementary Note: Data Acceptance Criteria, meaning that all 244 are valid.

Microsphere dilution and fluorescein dilution: For both of these protocols, a dilution level is considered locally valid if the measured value does not appear to be either saturated high or low. High saturation is determined by lack of sufficient slope from the prior level, here set to be at least 1.5x, and low saturation by indistinguishability from the blank replicates, here set to be anything <2 blank standard deviations above the mean blank. The valid range of dilution levels is then taken to be the longest continuous sequence of locally valid dilution levels, and the calibration set considered valid overall if this range has at least three valid dilution levels.

For microsphere dilution, of the 244 data sets, 235 are valid and 9 are not valid —one due to being entirely low saturated, the others having inconsistent slopes indicative of pipetting problems. Supplementary Fig. 2 Length of Valid Sequence(a) shows that most microsphere dilution data sets have the majority of dilution levels valid, but that only about one-tenth are without saturation issues.

For fluorescein dilution, of the 244 data sets, 243 are valid and 1 is not valid, having an inconsistent slope indicative of pipetting problems. Supplementary Fig. 2 Length of Valid Sequence(b) shows that the vast majority of fluorescein dilution data sets are without any saturation issues.

Note that in both cases, changing the required number of value dilution levels down to 2 or up to 4 would have little effect on the number of data sets included, adding 7 or removing 8 for microspheres and adding or removing only 1 for fluorescein.

## Unit scaling factor computation

*CFU*. The scaling factor $S_c$ relating CFU/ML to Abs600 is computed as follows:

$$S_{c,i} = \mu(C_i) * \delta_i \tag{1}$$

where $\mu(C_i)$ is the mean number of colonies for dilution level $i$ and $\delta_i$ is the dilution fold for level $i$. For the specific protocol used, there are three effective dilution factors, 1.6e5, 1.6e6, and 1.6e7 (including a 2-fold conversion between 200 and 100 μl volumes).

The overall scaling factor $S_c$ for each data set is then taken to be:

$$S_c = \left\{ S_{c,i} \middle| \frac{\sigma(C_i)}{\mu(C_i)} = \min_i \frac{\sigma(C_i)}{\mu(C_i)} \right\} \tag{2}$$

i.e., the scaling factor for the valid level with the lowest coefficient of variation, where $\sigma(C_i)$ is the standard deviation in the number of colonies for dilution level $i$.

The residuals for this fit are then $S_{c,i}/S_c$ for all other valid levels.

*LUDOX/water*. The scaling factor $S_l$ relating standard OD to Abs600 is computed as follow:

$$S_l = \frac{R}{\mu(L) - \mu(W)} \tag{3}$$

where $R$ is the measured reference OD in a standard cuvette (in this case 0.063 for LUDOX CL-X), $\mu(L)$ is the mean Abs600 for LUDOX CL-X samples and $\mu(W)$ is the mean Abs600 for water samples.

No residuals can be computed for this fit, because there are two measurements and two degrees of freedom.

*Microsphere dilution and fluorescein dilution*. The scaling factors $S_m$ relating microsphere count to Abs600 and $S_f$ for relating molecules of fluorescein to arbitrary fluorescent units are both computed in the same way. These are transformed into scaling factors in two ways, either as the mean conversion factor $S_\mu$ or as one parameter of a fit to a model of systematic pipetting error $S_p$.

*Mean conversion factor*: If we ignore pipetting error, then the model for serial dilution has an initial population of calibrant $p_0$ that is diluted $n$ times by a factor of $\alpha$ at each dilution, such that the expected population of calibrant for the $i$th dilution level is:

$$p_i = p_0(1 - \alpha)\alpha^{i-1} \tag{4}$$

In the case of the specific protocols used here, $\alpha = 0.5$. For the microsphere dilution protocol used, $p_0 = 3.00e8$ microspheres, while for the fluorescein dilution protocol used, $p_0 = 6.02e14$ molecules of fluorescein.

The local conversion factor $S_i$ for the $i$th dilution is then:

$$S_i = \frac{p_i}{\mu(O_i) - \mu(B)} \tag{5}$$

where $\mu(O_i)$ is the mean of the observed values for the $i$th dilution level and $\mu(B)$ is the mean observed value for the blanks.

The mean conversion factor is thus

$$S_\mu = \mu(\{S_i | i \text{ is a valid dilution level}\}) \tag{6}$$

i.e., the mean over local conversion factors for valid dilution levels.

The residuals for this fit are then $S_i/S_\mu$ for all valid levels.

*Systematic pipetting error model*: The model for systematic pipetting error modifies the intended dilution factor $\alpha$ with the addition of an unknown bias $\beta$, such that the expected biased population $b_i$ for the $i$th dilution level is:

$$b_i = p_0(1 - \alpha - \beta)(\alpha + \beta)^{i-1} \tag{7}$$

We then simultaneously fit $\beta$ and the scaling factor $S_p$ to minimize the sum squared error over all valid dilution levels:

$$\epsilon = \sum_i \left| \log \left( \frac{b_i}{S_p \cdot (\mu(O_i) - \mu(B))} \right) \right|^2 \tag{8}$$

where $\epsilon$ is sum squared error of the fit.

The residuals for this fit are then the absolute ratio of fit-predicted to observed net mean $\frac{b_i/S_p}{\mu(O_i) - \mu(B)}$ for all valid levels.

*Application to* E. coli *data*. The Abs600 and fluorescence a.u. data from *E. coli* samples are converted into calibrated units by subtracting the mean blank media values for Abs600 and fluorescence a.u., then multiplying by the corresponding scaling factors for fluorescein and Abs600.

**Criteria for valid *E. coli* data**. For analysis of *E. coli* culture measurements, a data set was only eligible to be included if both its fluorescence calibration and selected OD calibration were above a certain quality threshold. The particular values used for the four calibration protocols were:

CFU: Coefficient of variation for best dilution level is <0.5.

LUDOX/water: Coefficient of variation for both LUDOX and water are <0.1.

Microsphere dilution: Systematic pipetting error has geometric mean absolute residual <1.1-fold.

Fluorescein dilution: Systematic pipetting error has geometric mean absolute residual <1.1-fold.

Measurements of the cellular controls were further used to exclude data sets with apparent problems in their protocol: those with a mean positive control value more than 3-fold different than the median mean positive control.

Finally, individual samples without sufficient growth were removed, that being defined as all that are either less than the 25% of the 75th percentile Abs600 measurement in the sample set or less than 2 media blank standard deviations above the mean media blank in the sample set.

**Flow cytometry data processing**. Flow cytometry data was processed using the TASBE Flow Analytics software package[15]. A unit conversion model from arbitrary units to MEFL was constructed per the recommended best practices of TASBE Flow Analytics for each data set using the bead sample and lot information provided by each team.

Gating was automatically determined using a two-dimensional Gaussian fit on the forward-scatter area and side-scatter area channels for the first negative control (Supplementary Fig. 3).

The same negative control was used to determine autofluorescence for background subtraction (Supplementary Fig. 4).

As only a single green fluorescent protein was used, there was no need for spectral compensation or color translation.

All teams submitted flow cytometry used standard SpheroTech Rainbow Calibration beads[12] for dye-based calibration to equivalent fluorescent molecules[16]. In particular, 16 teams used RCP-30-5A beads (various lot numbers) and 1 team used URCP-38-2K beads, and conversion from arbitrary units to MEFL was computed using the peak-to-intensity values provided for each lot. Examples are provided below (Supplementary Figs. 5 and 6).

This color model was then applied to each sample to filter events and convert GFP measurements from arbitrary units to MEFL, and geometric mean and standard deviation computed for the filtered collection of events.

**Statistics and reproducibility**. As reproducibility is the main subject of this study, see the Results section above for its full presentation. In addition to the discussion of statistical analyses in the Results section, we note the following details of statistical analyses:

Coefficient of variation (CV) is computed per its definition, as the ratio of the standard deviation to the mean.

Fluorescence values are analyzed geometric mean and geometric standard deviation, rather than the more typical arithmetic statistics, due to the typical log-normal distribution of gene expression[13].

Data analysis was performed with Matlab.

**Reporting summary**. Further information on research design is available in the Nature Research Reporting Summary linked to this article.

## Data availability

All data generated or analyzed during this study are included in this published article (and its Supplementary Information files).

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

## Acknowledgements

Partial support for this work was provided by NSF Expeditions in Computing Program Award #1522074 as part of the Living Computing Project, and by the Engineering and Physical Sciences Research Council [EP/R034915/1] and EU H2020 [820699]. This document does not contain technology or technical data controlled under either the U.S. International Traffic in Arms Regulations or the U.S. Export Administration Regulations.

## Author contributions

Conceptualization: J.B., N.G.F., T.H-A., V.S.-1, G.S.B., R.B-T., M.G., D.K., J.M., and C.T.W. Data curation: J.B., N.G.F., T.H-A., and V.S.-1. Formal analysis: J.B. Investigation: Experimental data gathered by iGEM Interlab Study Contributors Methodology: J.B., N.G.F., T.H-A., V.S.-1, G.S.B., R.B-T., M.G., D.K., J.M., V.S.-2, A.S., and C.T.W. Project administration: J.B., N.G.F., and T.H-A. Resources: T.H-A., V.S.-1, and A.S. Software: J.B. Writing (original draft): J.B. and N.G.F. Writing (review & editing): J.B., N.G.F., T.H-A., G.S.B., J.M., C.T.W., and V.S.-2.

## Competing interests

The authors declare no competing interests.

## Additional information

## iGEM Interlab Study Contributors

### Aachen

Meryem Pehlivan[9] & Biel Badia Roige[9]

### Aalto-Helsinki

Tiu Aarnio[10,11], Samu Kivisto[10,11], Jessica Koski[10,11], Leevi Lehtonen[10,11], Denise Pezzutto[10,11] & Pauliina Rautanen[10,11]

### AHUT_China

Weixin Bian[12], Zhiyuan Hu[12], Zhihao Liu[12], Zi Liu[12], Liang Ma[12], Luyao Pan[12], Zichen Qin[12], Huichao Wang[12], Xiangxuan Wang[12], Hao Xu[12] & Xia Xu[12]

### Aix-Marseille

Yorgo El Moubayed[13]

## ASTWS-China
Shan Dong[14], Choco Fang[14], Hanker He[14], Henry He[14], Fangliang Huang[14], Ruyi Shi[14], Cassie Tang[14], Christian Tang[14], Shirly Xu[14] & Calvin Yan[14]

## Athens
Natalia Bartzoka[15,16], Eleni Kanata[15,16], Maria Kapsokefalou[15,16], Xanthi-Leda Katopodi[15,16], Eleni Kostadima[15,16], Ioannis V. Kostopoulos[15,16], Stylianos Kotzastratis[15,16], Antonios E. Koutelidakis[15,16], Vasilios Krokos[15,16], Maria Litsa[15,16], Ioannis Ntekas[15,16], Panagiotis Spatharas[15,16], Ourania E. Tsitsilonis[15,16] & Anastasia Zerva[15,16]

## Austin_LASA
Vidhya Annem[17], Eli Cone[17], Noel Elias[17], Shreya Gupta[17], Kendrick Lam[17] & Anna Tutuianu[17]

## Austin_UTexas
Dennis M. Mishler[18] & Bibiana Toro[18]

## Baltimore_BioCrew
Akinwumi Akinfenwa[19], Frank Burns[19], Heydy Herbert[19], Melissa Jones[19], Sarah Laun[19], Shikei Morrison[19] & Zion Smith[19]

## BCU
Zhao Peng[20] & Zhou Ziwei[20]

## BFSUICC-China
Rui Deng[21], Yilin Huang[21], Tingyue Li[21], Yingqi Ma[21], Zhiyuan Shen[21], Chenxi Wang[21], Yuyao Wang[21] & Tianyan Zhao[21]

## BGIC-Global
Yusen Lang[22], Yuteng Liang[22], Xueyao Wang[22] & Yi Wu[22]

## BGU_Israel
Dror Aizik[23], Sagi Angel[23], Einan Farhi[23], Nitzan Keidar[23], Eden Oser[23] & Mor Pasi[23]

## Bielefeld-CeBiTec
Jorn Kalinowski[24], Matthias Otto[24] & Johannes Ruhnau[24]

## Bilkent-UNAMBG
Hande Cubukcu[25], Mehmet Ali Hoskan[25] & Ilayda Senyuz[25]

## BioIQS-Barcelona
Jordi Chi[26], Antoni Planas Sauter[26] & Magda Faijes Simona[26]

## BioMarvel
Sumin Byun[27], Sungwoo Cho[27], Goeun Kim[27], Yeonjae Lee[27], Sangwu Lim[27] & Hanyeol Yang[27]

## BIT
Tian Xin[28], Zhang Yaxi[28] & Peng Zhao[28]

**BIT-China**

Weitang Han[28], Fa He[28], Yuna He[28], Nuonan Li[28] & Xiaofan Luo[28]

**BJRS_China**

Cheng Boxuan[29], Hu Jiaqi[29], Yang Liangjian[29], Li Wanji[29], Chen Xinguang[29] & Liu Xinyu[29]

**BNDS_CHINA**

Zishi Wu[30], Yukun Xi[30], Xilin Yang[30], Yuchen Yang[30], Zhuoyi Yang[30], Yihao Zhang[30] & Yuezhang Zhou[30]

**BNU-China**

Yue Peng[31], Liu Yadi[31], Shaobo Yang[31], Jiang Yuanxu[31] & Kecheng Zhang[31]

**BOKU-Vienna**

Doris Abraham[32] & Theresa Heger[32]

**BostonU**

Cass Leach[33], Kevin Lorch[33] & Linda Luo[33]

**British_Columbia**

Alex Gaudi[34], Anthony Ho[34], Morris Huang[34], Christine Kim[34], Luxcia Kugathasan[34], Kevin Lam[34], Catherine Pan[34], Ariel Qi[34] & Cathy Yan[34]

**Calgary**

Kaitlin Schaaf[35] & Cassandra Sillner[35]

**Cardiff_Wales**

Ryan Coates[36], Hannah Elliott[36], Emily Heath[36], Evie McShane[36], Geraint Parry[36], Ali Tariq[36] & Sophie Thomas[36]

**CCU_Taiwan**

Ching-Wei Chen[37], Yu-Hong Cheng[37], Chia-Wei Hsu[37], Chin-Hsuan Liao[37], Wei-Ting Liu[37], Yu-Cheng Tang[37], Yu-Hsin Tang[37] & Zon En Yang[37]

**CDHSU-CHINA**

Liu Jian[38,39], Caidian Li[38,39], Chenyi Lin[38,39], Guozheng Ran[38,39], Zhouyan Run[38,39], Weiyu Ting[38,39], Zhangxiang Yong[38,39] & Liuhong Yu[38,39]

**Chalmers-Gothenburg**

Andrea Clausen Lind[40], Axel Norberg[40], Amanda Olmin[40], Jacob Sjolin[40], Agnes Torell[40], Cecilia Trivellin[40], Francisco Zorrilla[40] & Philip Gorter de Vries[40]

**CIEI-BJ**

Haolun Cheng[41], Jiarong Peng[41] & Zhenyu Xiong[41]

**CMUQ**

Dina Altarawneh[42], Sayeda Sakina Amir[42], Sondoss Hassan[42] & Annette Vincent[42]

**CO_Mines**

Ben Costa[43], Isabella Gallegos[43], Mitch Hale[43], Matt Sonnier[43] & Kathleen Whalen[43]

**ColumbiaNYC**

Max Elikan[44], Sean Kim[44] & Jaewon You[44]

**Cornell**

Rahul Rambhatla[45] & Ashwin Viswanathan[45]

**CPU_CHINA**

Hong Tian[46], Huandi Xu[46], Wanli Zhang[46] & Shuyao Zhou[46]

**CSU_CHINA**

Liu Jiamiao[47] & Xiao Jiaqi[47]

**CSU_Fort_Collins**

Darilyn Craw[48], Marley Goetz[48], Neil Rettedal[48] & Hayden Yarbrough[48]

**Delgado-Ivy-Marin**

Christopher Ahlgren[49], Brett Guadagnino[49], James Guenther[49] & Juilanne Huynh[49]

**DLUT_China**

Zhien He[49], Huan Liu[49], Yuansheng Liu[49], Mingbo Qu[49], Li Song[49], Chao Yang[49], Jun Yang[49], Xianqi Yin[49], Yuanzhen Zhang[49], Jianan Zhou[49] & Lihan Zi[49]

**DLUT_China_B**

Zhu Jinyu[49], Xu Kang[49], Peng Xilei[49], Han Xue[49] & Shu Xun[49]

**DNHS_SanDiego**

Priyanka Babu[50], Arushi Dogra[50] & Pranav Thokachichu[50]

**DTU-Denmark**

David Faurdal[8], Joen Haahr Jensen[8], Jacob Mejlsted[8], Lina Nielsen[8] & Tenna Rasmussen[8]

**Duesseldorf**

Jennifer Denter[51], Kai Husnatter[51] & Ylenia Longo[51]

**Ecuador**

Juan Carlos Luzuriaga[52], Eduardo Moncayo[52], Natalia Torres Moreira[52] & Jennifer Tapia[52]

**ECUST**

Tang Dingyue[53], Zhao Jingjing[53], Xu Wenhao[53], Teng Xinyu[53] & Hong Xiujing[53]

**Edinburgh_OG**

Jackson DeKloe[54]

**Edinburgh_UG**

Ben Astles[54], Ugne Baronaite[54] & Inga Grazulyte[54]

**Emory**

Michael Hwang[55] & Yibo Pang[55]

**EPFL**

Michael Andrew Crone[56], Reza Hosseini[56], Moustafa Houmani[56], Daniel Zadeh[56] & Violetta Zanotti[56]

**ETH_Zurich**

Oliver Andreas Baltensperger[57], Eline Yafele Bijman[57], Elisa Garulli[57], Jan Lukas Krusemann[57], Adriano Martinelli[57], Antonio Martinez[57] & Tobias Vornholt[57]

**Evry_Paris-Saclay**

Monteil Camille[58] & Ahavi Paul[58]

**Exeter**

Emily Browne[59], Daniel Barber James Gilman[59], Amy Hewitt[59], Sophie Hodson[59], Ingebjorg Holmedal[59], Fiona Kennedy[59] & Juliana Sackey[59]

**FAU_Erlangen**

Selina Beck[60], Franziska Eidloth[60], Markus Imgold[60], Anna Matheis[60], Tanja Meerbrei[60], David Ruscher[60] & Marco Schaeftlein[60]

**FJNU-China**

Zhu Hanrong[61]

**Fudan**

Mitchell Wan[62]

**Fudan-CHINA**

Leijie Dai[62], Kaifeng Jin[62], Sihan Wang[62], Xin Wang[62], Yi Wang[62], Yifan Wang[62], Chenhai Wu[62], Zixuan Zhang[62] & Yineng Zhou[62]

**GDSYZX**

Liu Xinyu[63] & Zeng Zirong[63]

**Georgia_State**

Rehmat Babar[64], Mathew Brewer[64], Christina Clodomir[64], Laura Das Neves[64], Amanda Iwuogo[64], Ari Jones[64], Cara Jones[64], Julia Kelly[64], Gloria Kim[64], Jessica Siemer[64] & Yash Yadav[64]

**Gifu**

Yuichiro Ikagawa[65], Tatsuki Isogai[65] & Ryo Niwa[65]

## GO_Paris-Saclay

Celine Aubry[58], William Briand[58], Annick Jacq[58], Sylvie Lautru[58], Britany Marta[58], Clemence Maupu[58], Xavier Ollessa-Daragon[58], Kenn Papadopoulo[58] & Mahnaz Sabeta Azad[58]

## GreatBay_China

Wei Kuangyi[66], Yao Xiu[66] & Chenghao Yang[66]

## Groningen

Aditya Iyer[67], Rianne Prins[67] & Phillip Yesley[67]

## GZHS-United

Fang Lichi[68] & Chen Zi Xuan[68]

## HAFS

Kyuhee Jo[69], Mikyung Park[69], Seunghyun Park[69] & Hojun Yoo[69]

## Hamburg

Nele Burckhardt[70], Lea Daniels[70], Bjarne Klopprogge[70], Dustin Kruger[70], Oda-Emilia Meyfarth[70], Lisa Putthoff[70] & Dominika Wawrzyniak[70]

## HBUT-China

Xinyi Hu[71] & Yunyi Wang[71]

## HebrewU

Lior Badash[72], Amichai Baichman-Kass[72], Alon Barshap[72], Yonatan Friedman[72], Eliya Milshtein[72] & Omri Vardi[72]

## HFLS_ZhejiangUnited

Shan Dong[73,74], Yining Gu[73,74], Yuanzhe Pei[73,74], Ruyi Shi[73,74], Fan Yang[73,74], Jinshu Yang[73,74] & Xueqian Zhu[73,74]

## HK_HCY_LFC

Lam Kai Ching[75,76], Law Hiu Ching[75,76], Ng Tsz Chun[75,76], Yu Man Hin[75,76], Lai Tsz Hong[75,76], Chan Wing Lam[75,76], Yiu Choi Lam[75,76], Cheah Matthew[75,76], Cheng Tsz Ngo[75,76], Yun Shuan[75,76], Chan Tsey Wan[75,76], Tsui Shing Yan[75,76], Chong Yuk Yee[75,76], Tam Chi Yu[75,76] & Yuen Wai Yu[75,76]

## HKJS_S

Chung Tsun Ho Anson[77], Lee Sze Choi[77], Cheung Man Chun[77], Chan Lok Hin[77], Wong Chung Hin[77], Ng Sze Ho[77], Leung Chung Yin Jay[77], Lai Man Wai Katherine[77], Wong Carol Kin-ning[77], Lee Hong Kiu[77], Cheng Chak Kong[77], Leung Chung Wai[77], Yeung Wing Yan[77], Wong Tsz Yeung[77] & Lee Ka Yin[77]

## Hong_Kong_HKU

Tsui Shing Yan Grace[78], Lam Kai Ching Joe[78], Ng Tsz Chun Kenneth[78] & Cheah Matthew Yun Shuan[78]

**Hong_Kong_HKUST**

Ferdinan Aldo[79], Chung Him Pang[79], Kam Pang So[79] & Hei Man Wong[79]

**Hong_Kong_JSS**

Lai Tsz Ching[80,81], Luk Hau Ching[80,81], Ip Ning Fung[80,81], Yam Shing Fung[80,81], Lee Chi Hong[80,81], Hsiu Ou Ning[80,81] & Jonathan Cheng Hon Sang[80,81]

**Hong_Kong-CUHK**

Yeung Hoi Lam Elsa[82], Chan Yick Hei[82], Lo Ho Sing[82] & Choi Seong Wang[82]

**HUBU-Wuhan**

Yiheng Gu[83], Ziyue Rong[83], Haoyue Song[83], Pengying Wang[83] & Yuefei Wang[83]

**HUST-China**

Yan Chen[84], Hao Qiu[84], Haotian Ren[84] & Ziyang Xiao[84]

**HZAU-China**

Heng Heng[85], Xichen Rao[85] & Ruonan Tian[85]

**ICT-Mumbai**

Shalini S. Deb[86], Yash Laxman Kamble[86], Ninad Kumbhojkar[86], Bhargav Patel[86], Supriya Prakash[86], Shamlan M. S. Reshamwala[86] & Poorva Taskar[86]

**IISc-Bangalore**

Gokul[87] & Adwaith B. Uday[87]

**IISER-Bhopal-India**

Anubhav Basu[88], Rishi Gandhi[88], Jatin Khaimani[88], Arundhati Khenwar[88], Sandeep Raut[88] & Tejas Somvanshi[88]

**IISER-Kolkata**

Diptatanu Das[89], Souvik Ghosh[89] & Hrishika Rai[89]

**IISER-Mohali**

Nithishwer Mouroug Anand[90], Ashwin Kumar Jainarayanan[90], Pranshu Kalson[90], Devang Haresh Liya[90], Vibhu Mishra[90], Sveekruth Sheshagiri Pai[90], Madhav Pitaliya[90], Yash Rana[90] & Ravineet Yadav[90]

**IIT_Delhi**

Neha Arora[91], Vasu Arora[91], Shubham Jain[91], Abhilash Patel[91], Saksham Sharma[91] & Priyanka Singh[91]

**IIT_Kanpur**

Anushya Goenka[92], Rishabh Jain[92], Aryaman Jha[92], Adarsh Kumar[92] & Abhinav Soni[92]

**IIT-Madras**

Sathvik Ananthakrishnan[93], Velvizhi Devi[93], Mohammed Faidh[93], Guhan Jayaraman[93], M. Sagar Kittur[93], Nitish R. Mahapatra[93], Sarvesh Menon[93], Anantha Barathi Muthukrishnan[93], B. P. Kailash[93], Burhanuddin Sabuwala[93], Mousami Shinde[93] & Sankalpa Venkatraghavan[93]

**Jiangnan_China**

Weijia Liu[94], Zhoudi Miao[94], Tian Wang[94], Yaling Wang[94] & Shuyan Zhang[94]

**Jilin_China**

Ruochen Chai[95], Yubin Ge[95], Ali Hou[95], Fangqi Liu[95], Xutong Liu[95], Jiangjiao Mao[95], Zihao Wang[95], Haimeng Yu[95], Hetian Yuan[95] & Yang Zhan[95]

**JMU_Wuerzburg**

Anna Ries[96] & Chiara Wolfbeisz[96]

**KAIT_JAPAN**

Toshihiro Kanaya[97], Yusuke Kawasaki[97], Tatuya Maruo[97], Yuya Mori[97] & Takehito Satoh[97]

**KCL_UK**

Anthony Chau[98], Wai Yan Chu[98], Anatoliy Markiv[98], Marcos Vega-Hazas Marti[98], Maria Jose Ramos Medina[98], Deeksha Raju[98] & Shubhankar Sinha[98]

**KUAS_Korea**

Youngeun Choi[99] & Bo Sun Ryu[99]

**Lambert_GA**

Gaurav Byagathvalli[100] & Ellie Kim[100]

**Leiden**

Marjolein Crooijmans[101], Jazzy de Waard[101] & Chiel van Amstel[101]

**Lethbridge**

Aubrey Demchuk[102], Travis Haight[102], Dong Ju Kim[102], Andrei Neda[102], Luc Roberts[102], Luke Saville[102], Reanna Takeyasu[102] & David Tobin[102]

**Lethbridge_HS**

Mina Akbary[103,104,105,106], Rebecca Avileli[103,104,105,106], Karen He[103,104,105,106], Aroma Pageni[103,104,105,106], Luke Saville[103,104,105,106], Dewuni De Silva[103,104,105,106], Nimaya De Silva[103,104,105,106], Kristi Turton[103,104,105,106], Michelle Wu[103,104,105,106] & Alice Zhang[103,104,105,106]

**Lubbock_TTU**

Benjamin Chavez[107], Paula Garavito[107], Michael Latham[107], Jeffrey Ptak[107] & Darron Tharp[107]

**Lund**

Nurul Izzati[108,109], Martin Jonsson[108,109], Nikol Labecka[108,109] & Sara Palo[108,109]

**Macquarie_Australia**

Renee Beale[110], Dominic Logel[110], Areti-Efremia Mellou[110] & Karl Myers[110]

## Madrid-OLM

Alejandro Alonso[111,112], Rodrigo Hernandez Cifuentes[111,112], Borja Sanchez Clemente[111,112], Gonzalo Saiz Gonzalo[111,112], Ivan Martin Hernandez[111,112], Laura Armero Hernandez[111,112], Francisco Javier Quero Lombardero[111,112], Domingo Marquina[111,112], Guillermo Fernandez Rodriguez[111,112] & Ignacio Albert Smet[111,112]

## Manchester

Tom Butterfield[113], Ed Deshmukh-Reeves[113], Namrata Gogineni[113], Sam Hemmings[113], Ismat Kabbara[113], Ieva Norvaisaite[113] & Ryan Smith[113]

## Marburg

Daniel Bauersachs[114], Benjamin Daniel[114], Rene Inckemann[114], Alexandra Seiffermann[114], Daniel Stukenberg[114] & Carl Weile[114]

## McGill

Valerian Clerc[115], Jacqueline Ha[115] & Stephanie Totten[115]

## McMaster

Thomas Chang[116], Carlene Jimenez[116] & Dhanyasri Maddiboina[116]

## METU_HS_Ankara

Beliz Leyla Acar[117], Evrim Elcin[117], Tugba Inanc[117], Gamze Kantas[117], Ceyhun Kayihan[117], Mert Secen[117], Gun Suer[117], Kutay Ucan[117] & Tunc Unal[117]

## Michigan

Matthew Fischer[118], Naveen Jasti[118] & Thomas Stewart[118]

## MichiganState

Sarah Caldwell[119], Jordan Lee[119] & Jessica Schultz[119]

## Mingdao

Ting-Chen Chang[120], Pei-Hong Chen[120], Yu-Hsuan Cheng[120], Yi-Hsuan Hsu[120] & Chan-yu Yeh[120]

## Minnesota

Zhipeng Ding[121], Zihao Li[121], Savannah Lockwood[121] & Katherine Quinn[121]

## Montpellier

Leo Carrillo[122], Maxime Heintze[122], Lea Meneu[122], Marie Peras[122] & Tamara Yehouessi[122]

## Munich

Keno Eilers[123,124], Elisabeth Falgenhauer[123,124], Wong Hoi Kiu[123,124], Julia Mayer[123,124], Julia Mueller[123,124], Sophie von Schoenberg[123,124], Dominic Schwarz[123,124] & Brigit Tunaj[123,124]

## Nanjing-China

Zhaoqing Hu[125], Yansong Huang[125] & Yuanyuan Li[125]

**NAU-CHINA**
Chengzhu Fang[126], Jiangyuan Liu[126], Yiheng Liu[126], Yaxuan Wu[126], Sheng Xu[126] & Long Yuan[126]

**NAWI_Graz**
Marco Edelmayer[127,128,129], Marlene Hiesinger[127,128,129], Sebastian Hofer[127,128,129], Birgit Krainer[127,128,129], Andreas Oswald[127,128,129], Dominik Strasser[127,128,129] & Andreas Zimmermann[127,128,129]

**NCHU_Taichung**
Yi-Cian Chen[130]

**NCTU_Formosa**
Yuan-Yao Chan[131], Yu-Ci Chang[131], Nian Ruei Deng[131], Chi-Yao Ku[131] & Meng-Zhan Lee[131]

**NEU_China_A**
Hailong Li[132], Zhaoyu Liu[132], Guowei Song[132], Yuening Xiang[132] & Hongfa Yan[132]

**NEU_China_B**
He Huanying[132], Jiang Qiaochu[132], Jiang Shengjuan[132] & Peng Yujie[132]

**Newcastle**
Matt Burridge[133], Kyle Standforth[133] & Sam Went[133]

**NJU-China**
Liang Chenxi[125], Wang Han[125], Zhang Qipeng[125], Li Yifan[125], Quan Yiming[125] & Pan Yutong[125]

**NKU_CHINA**
Senhao Kou[134] & Lin Luan[134]

**Northwestern**
Umut Akova[135], Liza Fitzgerald[135], Bon Ikwuagwu[135], Michael Johnson[135], Jacob Kurian[135] & Christian Throsberg[135]

**Nottingham**
Lucy Allen[136], Christopher Humphreys[136], Daniel Partridge[136], Michaella Whittle[136] & Nemira Zilinskaite[136]

**NPU-China**
Meixuan Lee[137], Weifeng Lin[137], Yuan Ma[137] & Kai Wang[137]

**NTHU_Formosa**
Hsuan Cheng[138], Shumei Chi[138], Yi-Chien Chuang[138], Ray Huang[138], LiangYu Ko[138] & Yu-Chun Lin[138]

**NTHU_Taiwan**
You-Yang Tsai[138], Cheng-Chieh Wang[138] & Kai-Chiang Yu[138]

**NTNU_Trondheim**
Hanna Nedreberg Burud[139], Carmen Chen[139], Anne Kristin Haralsvik[139], Adrian Marinovic[139], Hege Hetland Pedersen[139], Amanda Sande[139] & Vanessa Solvang[139]

**NTU-Singapore**
Shaw Kar Ming[140] & Albert Praditya[140]

**NU_Kazakhstan**
Aiganym Abduraimova[141], Ayagoz Meirkhanova[141], Assel Mukhanova[141] & Tomiris Mulikova[141]

**NUDT_CHINA**
Yanchen Gou[142], Chenyu Lu[142], Jiaxin Ma[142] & Chushu Zhu[142]

**NUS_Singapore-A**
Leow Chung Yong Aaron[143], Tvarita Shivakumar Iyer[143], Wu Jiacheng[143], Yan Ping Lim[143], Beatrix Tung Xue Lin[143], Aaron Ramzeen[143] & Nur Liyana Binte Ayub Ow Yong[143]

**NUS_Singapore-Sci**
Yah Tse Sabrina Chua[143], Yuhui Deborah Fong[143], Menglan He[143] & Li Yang Tan[143]

**NWU-China**
Zhang Jiahe[144], Li Mingge[144], Li Nianlong[144], Li Yueyi[144] & Cheng Yuhan[144]

**NYMU-Taipei**
Annabel Chang[145], Chih-Chiang Chen[145], Ryan Chou[145], Jude Clapper[145], Evelyn Lai[145], Yasmin Lin[145], Kelsey Wang[145] & Jake Yang[145]

**NYU_Abu_Dhabi**
Mariam Anwar[146], Ibrahim Chehade[146], Imtiyaz Hariyani[146], Sion Hau[146], Ashley Isaac[146], Laura Karpauskaite[146], Mazin Magzoub[146], Daniel Obaji[146], Yong Rafael Song[146] & Yejie Yun[146]

**OUC-China**
Kai Sun[147] & Yunqian Zhang[147]

**Oxford**
Eleanor Beard[148], Laurel Constanti Crosby[148], Nicolas Delalez[148], Arman Karshenas[148], Adrian Kozhevnikov[148], Jhanna Kryukova[148], Karandip Saini[148], Jon Stocks[148], Bhuvana Sudarshan[148], Max Taylor[148], George Wadhams[148] & Joe Windo[148]

**Paris_Bettencourt**
Annissa Ameziane[149], Darshak Bhatt[149], Alexis Casas[149], Antoine Levrier[149], Ana Santos[149], Nympha Elisa M. Sia[149] & Edwin Wintermute[149]

**Pasteur_Paris**

Alice Dejoux[150], Deshmukh Gopaul[150], Lea Guerassimoff[150], Samuel Jaoui[150], Manon Madelenat[150] & Serena Petracchini[150]

**Peking**

Fu Cai[151], Yang Jianzhao[151], Shi Shuyu[151], Li Tairan[151], Li Xin[151], Lin Yongjie[151] & Huang Zhecheng[151]

**Pittsburgh**

Evan Becker[152], Matthew Greenwald[152], Vivian Hu[152], Tucker Pavelek[152], Elizabeth Pinto[152] & Zemeng Wei[152]

**Purdue**

Zachary Burgland[153], Janice Chan[153], Julianne Dejoie[153], Kevin Fitzgerald[153], Zach Hartley[153], Moiz Rasheed[153] & Makayla Schacht[153]

**Queens_Canada**

Maddison Gahagan[154], Ellis Kelly[154] & Elisha Krauss[154]

**RDFZ-China**

Yuze Cao[155], Yishen Shen[155], Xuan Wang[155], Hanning Xu[155] & Jianxiang Zhang[155]

**REC-CHENNAI**

Priyanka Chandramouli[156], Amal Jude Ashwin F[156], Srimathi Jayaraman[156], Marcia Smiti Jude[156], Vignesh Kumar[156], Hema Lekshmi[156], R. Preetha[156], Khadija Rashid[156], S. Deepak Kumar[156] & B. S. Mohan Kumar[156]

**Rheda_Bielefeld**

Leon Michael Barrat[157,158], Jil-Sophie Dissmann[157,158], Jorn Kalinowski[157,158], Matthias Otto[157,158], Johannes Ruhnau[157,158], Fynn Stuhlweissenburg[157,158] & Elisa Ueding[157,158]

**RHIT**

Ariel Bohner[159], Brittany Clark[159], Emilie Deibel[159], Liz Klaas[159], Kaylee Pate[159] & Elisa Weber[159]

**Rice**

Katherine Cohen[160], Anna Guseva[160], Stefanie King[160] & Soohyun Yoon[160]

**Ruia-Mumbai**

Sanika Ambre[161], Shilpa Bhowmick[161], Nishtha Pange[161], Komal Parab[161], Vainav Patel[161], Mitali Patil[161], Aishwarya Rajurkar[161], Mayuri Rege[161], Maithili Sawant[161], Shrutika Sawant[161] & Anjali Vaidya[161]

**SBS_SH_112144**

Peicheng Ji[162,163], Fang Luo[162,163], Guanghui Ma[162,163], Xin Xu[162,163], Jiacheng Yin[162,163], Yinchi Zhou[162,163] & Ke Zhu[162,163]

**SCAU-China**

Yaohua Huang[164], Yinpin Huang[164], Jiadong Li[164], Xuecheng Li[164], Hao Wang[164], Ken Wang[164], Wei Wang[164], Xinyu Zhang[164] & Jiahua Zou[164]

**SCU-China**

Minyue Bao[165], Han Kang[165], Xiaolong Liu[165], Yibing Tao[165], Ziru Wang[165], Fuqiang Yang[165], Tianyi Zhang[165] & Yanling Zhong[165]

**SCUT_ChinaB**

Jiezheng Liu[166], Jingang Liu[166], Lingling Ma[166], Xubo Niu[166], Ling Qian[166], Li Wang[166], Qingyan Yan[166] & Nannan Zhao[166]

**SCUT-ChinaA**

Weixuan Chen[166] & Yuxin Zhou[166]

**SDU-CHINA**

Junyang Chen[167]

**SFLS_Shenzhen**

Junyao Hao[168], Zhang HuaYue[168], Peilin Li[168], Yifei Pei[168], Jingting Qu[168], Raven Wang[168], Xinyue Wang[168], Kangjie Wu[168], Yuxuan Wu[168], Meredith Xiang[168], Leyi Yang[168], Zisang Yang[168] & Li Zhaoting[168]

**ShanghaiTech**

Wenhan Fu[169], Zonghao Li[169], Weiyi Tang[169] & Kaida Zhang[169]

**SHSBNU_China**

Haocong Li[170], Xuze Shao[170], Chuyi Yang[170], Yuanhong Zeng[170] & Yanjun Zhou[170]

**SHSID_China**

Shangzhi Dong[171], Younji Jung[171], Sophie Ruojia Li[171], Tingting Li[171] & Jiacheng Yu[171]

**SHSU_China**

Shangzhi Dong[172,173,174], Tingting Li[172,173,174], Xinyi Miao[172,173,174] & Sibo Wang[172,173,174]

**SIAT-SCIE**

Yiming Ding[175], Jiaxi Huang[175], Yuqi Li[175], Ting Sun[175], Qinghe Tian[175], Mengxuan Wu[175], Jinming Xing[175], Xin Xiong[175], Yining Yan[175], Qiu Yihang[175], Jige Zhang[175], Yi Zhou[175] & Zhiyu Zhou[175]

**SJTU-BioX-Shanghai**

Zhuoyang Chen[176], Peixiang He[176], Yirui Hong[176], Chia-Yi Hsiao[176], Zhihan Liang[176], Zhixiang Liu[176], Yuncong Ran[176], Shiyu Sun[176] & Ruoyu Xia[176]

**SKLMT-China**

Dongyang Dong[177] & Wenxue Zhao[177]

**SMMU-China**

Miao Hu[178], Shi Hu[178], Wei Shi[178], Shulun[178], Han Yan[178] & Yusheng Ye[178]

**SMS_Shenzhen**

Yiquan Hong[179], Yuyao Pan[179], Yiran Song[179], Jinhan Zhang[179] & Yihang Zhao[179]

**Sorbonne_U_Paris**

Dounia Chater[180], Asmaa Foda[180], Yanyan Li[180], Ursula Saade[180] & Victor Sayous[180]

**SSHS-Shenzhen**

Yilin Mo[181], Wenan Ren[181] & Chenxu Zeng[181]

**SSTi-SZGD**

Yixin Cao[182]

**St_Andrews**

Clarissa Czekster[183], Izzy Dunstan[183], Simon Powis[183], Bethany Reaney[183], Eva Snaith[183] & Cam Young[183]

**Stanford**

Eva Frankel[184], Eleanor Glockner[184] & Isaac Justice[184]

**Stanford-Brown-RISD**

Santosh Murugan[184,185,186] & Leo Penny[184,185,186]

**Stockholm**

Chrismar Garcia[187,188,189] & Stamatina Rentouli[187,188,189]

**Stony_Brook**

Priya Aggarwal[190], Stephanie Budhan[190], Woody Chiang[190], Dominika Kwasniak[190], Karthik Ledalla[190], Matthew Lee[190], Natalie Lo[190], Matthew Mullin[190], Lin Yu Pan[190], Jennifer Rakhimov[190], Robert Ruzic[190], Manvi Shah[190], Lukas Velikov[190] & Sara Vincent[190]

**Stuttgart**

Philip Horz[191], Nadine Kuebler[191] & Jan Notheisen[191]

**SUIS_Shanghai**

David Doyle[192], Jiajun Gu[192], Wenyue Hu[192] & Shuting Yang[192]

**SYSU-CHINA**

Tao Kehan[193], Gao Menghan[193] & Mao Xiaowen[193]

**SYSU-Software**

Yifei Chen[193], Ziqi Kang[193] & Haochen Ni[193]

**SZU-China**

Junyu Chen[194], Lindong He[194], Mingyue Luo[194] & Jiaqi Tang[194]

**Tacoma_RAINmakers**

Kira Boyce[195], James Lee[195], Michael Martin[195], Judy Van Nguyen[195] & Leon Wan[195]

**Tartu_TUIT**

Artur Astapenka[196], Turan Badalli[196], Irina Borovko[196], Nadezhda Chulkova[196], Ilona Faustova[196], Anastasia Kolosova[196], Mart Loog[196], Artemi Maljavin[196], Frida Matiyevskaya[196] & Vladislav Tuzov[196]

**TAS_Taipei**

Catherine Chang[197], Ryan Chou[197], Jude Clapper[197], Tim Ho[197], Yi Da Hsieh[197], Evelyn Lai[197], Leona Tsai[197], Kelsey Wang[197] & Justin Wu[197]

**Tec-Chihuahua**

Viana Isabel Perez Dominguez[198], Cesar Ibrahym Rodriguez Fernandez[198], Daniela Olono Fierro[198], Anna Karen Aguilar Nunez[198], Jose Pablo Rascon Perez[198], Mario Loya Rivera[198], Cynthia Lizeth Gonzalez Trevizo[198] & Maria Antonia Luna Velasco[198]

**Tec-Monterrey**

Carlos Javier Cordero Oropeza[199], Adrian Federico Hernandez Mendoza[199], Jose Arnulfo Juarez Figueroa[199], Luis Mario Leal[199], Samantha Ayde Pena Benavides[199], Victor Javier Robledo Martinez[199], Adriana Lizeth Rubio Aguirre[199], Andres Benjamin Sanchez Alvarado[199], Margarita Sofia Calixto Solano[199], Nora Esther Torres Castillo[199], Alejandro Robles Zamora[199] & Esteban de la Pena Thevenet[199]

**TecCEM**

Karla Soto Blas[200], Ana Laura Torres Huerta[200] & Armando Cortes Resendiz[200]

**TecMonterrey_GDL**

Frida Cruz[201], Fernanda Diaz[201], Diego Espinoza[201], Ana Cristina Figueroa[201], Ana Cecilia Luque[201], Roberto Portillo[201], Carolina Senes[201], Diana Tamayo[201] & Mariano Del Toro[201]

**Thessaloniki**

Ioannis Alexopoulos[202], Alexandros Dimitriou Giannopoulos[202], Yvoni Giannoula[202] & Grigorios Kyrpizidis[202]

**Tongji_China**

Ma Xinyue[203], Chen Xirui[203] & Song Zhiwei[203]

**Toronto**

Nina Adler[204], Amalia Caballero[204], Carla Hamady[204], Ahmed Ibrahim[204], Jasmeen Parmar[204], Tashi Rastogi[204] & Jindian Yang[204]

**Toulouse-INSA-UPS**

Jean Delhomme[205,206], Anthony Henras[205,206], Stephanie Heux[205,206], Yves Romeo[205,206], Marion Toanen[205,206], Camille Wagner[205,206] & Paul Zanoni[205,206]

**TU_Darmstadt**

Thea Lotz[207], Elena Nickels[207], Beatrix Suss[207], Heribert Warzecha[207] & Jennifer Zimmermann[207]

**TU-Eindhoven**
Emilien Dubuc[208], Bruno Eijkens[208], Sander Keij[208], Simone Twisk[208], Mick Verhagen[208] & Maxime van den Oetelaar[208]

**TUDelft**
Alexander Armstrong[209], Nicole Bennis[209], Susan Bouwmeester[209], Lisa Buller[209], Kavish Kohabir[209], Monique de Leeuw[209], Venda Mangkusaputra[209], Jard Mattens[209], Janine Nijenhuis[209], Timmy Paez[209], Lisbeth Schmidtchen[209] & Gemma van der Voort[209]

**TUST_China**
Gao Ge[210], Xu Haoran[210] & Li Xiaojin[210]

**UAlberta**
Ejouan Agena[211], Ethan Agena[211], Scott Bath[211], Robert Campbell[211], Rochelin Dalangin[211], Anna Kim[211], Dominic Sauvageau[211] & Irene Shkolnikov[211]

**UC_Davis**
Daniel Graves[212], Jacob Lang[212], Jolee Nieberding-Swanberg[212], Achala Rao[212], Ares Torres[212] & Andrew Yao[212]

**UC_San_Diego**
Anser Abbas[213] & Claire Luo[213]

**UCAS-China**
Xu Zepeng[214] & Zhao Ziyi[214]

**UChicago**
Janice Chen[215], Cian Colgan[215], Steve Dvorkin[215], Rachael Filzen[215], Varun Patel[215], Allison Scott[215] & Patricia Zulueta[215]

**UChile_Biotec**
Joaquin Acosta[216], Lucas Araya[216], Francisco Chavez[216], Sebastian Farias[216], Delia Garrido[216], Andres Marcoleta[216], Felipe Munoz[216] & Paula Rivas[216]

**UCL**
Noelle Colant[217], Catherine Fan[217], Stefanie Frank[217], Jacopo Gabrielli[217], Paola Handal[217], Vitor Pinheiro[217], Stefanie Santamaria[217], Shamal Withanage[217] & Fang Xue[217]

**UCLouvain**
Antoine Gerard[218], Marine Lefevre[218], Fiona Milano[218], Nina De Sousa Oliveira[218], Mathieu Parmentier[218] & Luca Rigon[218]

**UConn**
Elizabeth Chamiec-Case[219], Ryan Chen[219], Peter Crowley[219], Shannon Doyle[219], Sricharan Kadimi[219] & Toni Vella[219]

**UCopenhagen**

Natthawut Adulyanukosol[220], Theodore A. Dusseaux[220], Victor Forman[220], Cecilie Hansen[220], Selma Kofoed[220], Simon Louis[220], Magnus Ronne Lykkegaard[220], Davide Mancinotti[220], Lasse Meyer[220], Stephanie Michelsen[220], Morten Raadam[220], Victoria Svaerke Rasmussen[220], Eirikur Andri Thormar[220], Attila Uslu[220] & Nattawut leelahakorn[220]

**UESTC-China**

Shizhi Ding[221], Changyu Li[221], Huishuang Tan[221], Yinsong Xu[221] & Jianzhe Yang[221]

**UFlorida**

Diego Gamoneda[222], Nicole Kantor[222], Lidimarie Trujillo-Rodriguez[222] & Matthew Turner[222]

**UGA**

Stephan George[223], Kelton McConnell[223] & Chynna Pollitt[223]

**UI_Indonesia**

Ihya Fakhrurizal Amin[224], Muhammad Ikhsan[224], Valdi Ven Japranata[224], Andrea Laurentius[224], Luthfian Aby Nurachman[224] & Muhammad Iqbal Adi Pratama[224]

**UiOslo_Norway**

Yvette Dirven[225], Lisa Frohlich[225], Dirk Linke[225], Verena Mertes[225], Rebekka Rekkedal Rolfsnes[225] & Athanasios Saragliadis[225]

**UIOWA**

Sandra Castillo[226], Sathivel Chinnathambi[226], Craig Ellermeier[226], Jennifer Farrell[226], Jan Fassler[226], Ernie Fuentes[226], Sean Ryan[226] & Edward Sander[226]

**UIUC_Illinois**

Amie Bott[227], Liam Healy[227], Pranathi Karumanchi[227], Alex Ruzicka[227] & Ziyu Wang[227]

**ULaval**

Gabriel Byatt[228], Philippe C. Despres[228], Alexandre Dube[228], Pascale Lemieux[228], Florian Lepetit[228], Louis-Andre Lortie[228] & Francois D. Rouleau[228]

**ULaVerne_Collab**

Seth Barrington[229], Cynthia Basulto[229], Sabrina Delgadillo[229], Karen De Leon[229], Micah Madrid[229], Catherosette Meas[229], Angelica Sabandal[229], Magaly Aguirre Sanchez[229], Jennifer Tsui[229] & Noble Woodward[229]

**UMaryland**

Rohith Battina[230], Jess Boyer[230], Arjun Cherupalla[230], Jason Chiang[230], Mary Heng[230], Collin Keating[230], Tommy Liang[230], Chun Kit Loke[230], Jacob Premo[230], Keerthana Srinivasan[230], John Starkel[230] & Daniel Zheng[230]

**UNebraska-Lincoln**

Gabe Astorino[231], Rachel Van Cott[231], Jintao Guo[231], Drew Kortus[231] & Wei Niu[231]

**Unesp_Brazil**

Paulo J. C. Freire[232], Danielle Biscaro Pedrolli[232], Nathan Vinicius Ribeiro[232], Bruna Fernandes Silva[232], Nadine Vaz Vanini[232], Mariana Santana da Mota[232] & Larissa de Souza Crispim[232]

**UNSW_Australia**

Tyler Chapman[233], Tobias Gaitt[233], Megan Jones[233] & Emily Watson[233]

**UPF_CRG_Barcelona**

Guillem Lopez-Grado[234,235] & Laura Sans[234,235]

**UPF_CRG_Barcelona**

Matilda Brink[236], Varshni Rajagopal[236] & Elin Ramstrom[236]

**US_AFRL_CarrollHS**

Anna Bete[237], Yazmin Camacho[237], Jonah Carter[237], Christina Davis[237], Jason Dong[237], Amy Ehrenworth[237], Michael Goodson[237], Chris Guptil[237], Max Herrmann[237], Chia Hung[237], Hayley Jesse[237], Rachel Krabacher[237], Dallas McDonald[237], Peter Menart[237], Travis O'Leary[237], Laura Polanka[237], Andrea Poole[237] & Vanessa Varaljay[237]

**USMA-West_Point**

Alana Appel[238], John Cave[238], Liz Huuki[238], Matt McDonough[238], Channah Mills[238], Alex Mitropoulos[238], James Pruneski[238] & Ken Wickiser[238]

**USP-Brazil**

Felipe Xavier Buson[239], Vinicius Flores[239], Guilherme Meira Lima[239] & Caio Gomes Tavares Rosa[239]

**UST_Beijing**

Guanke Bao[240], Haitao Dong[240], Zhi Luo[240] & Jiarong Peng[240]

**USTC**

Yongyan An[241], Cheng Cheng[241], Zhenyu Jiang[241], Linzhen Kong[241], Chenfei Luo[241], Liudong Luo[241], Yingying Shi[241], Erting Tang[241], Ping Wang[241], Yuyang Wang[241], Guiyang Xu[241], Wenfei Yu[241], Bonan Zhang[241] & Qian Zhang[241]

**UT-Knoxville**

David Garcia[242], Nannan Jiang[242], Brandon Kristy[242], Ralph Laurel[242], Karl Leitner[242], Frank Loeffler[242], Steven Ripp[242] & Morgan Street[242]

**Utrecht**

Khadija Amheine[243], Felix Bindt[243], Meine Boer[243], Mike Boxem[243], Jolijn Govers[243], Seino Jongkees[243], Lorenzo Pattiradjawane[243], Pim Swart[243], Helen Tsang[243], Floor de Graaf[243], Marjolijn ten Dam[243] & Franca van Heijningen[243]

**Valencia_UPV**

Yadira Boada[244] & Alejandro Vignoni[244]

**Vilnius-Lithuania**

Valentas Brasas[245], Aukse Gaizauskaite[245], Gabrielius Jakutis[245], Simas Jasiunas[245], Ieva Juskaite[245], Justas Ritmejeris[245], Dovydas Vaitkus[245], Tomas Venclovas[245], Kornelija Vitkute[245], Hanna Yeliseyeva[245], Kristina Zukauskaite[245] & Justina Zvirblyte[245]

**Vilnius-Lithuania-OG**

Laurynas Karpus[245], Ignas Mazelis[245] & Irmantas Rokaitis[245]

**Virginia**

Ngozi D. Akingbesote[246], Dylan Culfogienis[246], William Huang[246] & Kevin Park[246]

**Warwick**

Janvi Ahuja[247], Christophe Corre[247], Gurpreet Dhaliwal[247], Rhys Evans[247], Kurt Hill[247], Olivor Holman[247], Alfonso Jaramillo[247], Alizah Khalid[247], Jack Lawrence[247], Laura Mansfield[247], James O'Brien[247], June Ong[247], Satya Prakash[247] & Jonny Whiteside[247]

**Washington**

Karl Anderson[248], Emily Chun[248], Grace Kim[248], Aerilynn Nguyen[248], Chemay Shola[248], Dorsa Toghani[248], Angel Wong[248], Joanne Wong[248] & Jay Yung[248]

**WashU_StLouis**

Elizabeth Johnson[249], Divangana Lahad[249], Kyle Nicholson[249], Havisha Pedamallu[249] & Cam Phelan[249]

**Waterloo**

Clara Fikry[250], Leah Fulton[250], Nicole Lassel[250], Dylan Perera[250], Marina Robin[250] & Nicolette Shaw[250]

**Westminster_UK**

Kyle Bowman[251], Sarah Coleman[251], Kristian Emilov[251], Camila Gaspar[251], Jenaagan Jenakendran[251], Sara Mubeen[251], Marko Obrvan[251] & Caroline Smith[251]

**WHU-China**

Tang Bo[252], Du Liaoqi[252], Chang Tianyi[252], Xing Yuan[252] & Qing Yue[252]

**William_and_Mary**

Stephanie Do[253], Xiangyi Fang[253], Ethan Jones[253], Jessica Laury[253], Wukun Liu[253], Adam Oliver[253], Lillian Parr[253], Margaret Saha[253], Chengwu Shen[253], Tinh Son[253], Julia Urban[253], Yashna Verma[253] & Hanmi Zhou[253]

**Worldshaper-Wuhan**

Shan Dong[254], Zhengguo Hao[254], Yi Kuang[254], Ting Liu[254] & Rui Zhou[254]

## WPI_Worcester

Beck Arruda[2], Natalie Farny[2], Mei Hao[2], Camille Pearce[2], Alex Rebello[2], Arth Sharma[2], Kylie Sumner[2] & Bailey Sweet[2]

## XJTLU-CHINA

Junliang Lin[255]

## XJTU-China

Du Mengtao[256], Fan Peiyao[256] & Fang Xinlei[256]

## XMU-China

Niangui Cai[257], Junhong Chen[257], Yousi Fu[257], Yunyun Hu[257], Ye Qiang[257], Qiupeng Wang[257], Ruofan Yang[257], Chen Yucheng[257] & Jiyang Zheng[257]

## Yale

Kevin Chang[258], Cecily Gao[258], Farren Isaacs[258], Kevin Li[258], Ricardo Moscoso[258], Jaymin Patel[258], Lauren Telesz[258] & Alice Tirad[258]

## ZJU-China

Qinhao Cao[259], Xinhua Feng[259], Yinjing Lu[259], Xianyin Zhang[259] & Xuanhao Zhou[259]

## ZJUT-China

Dongchang Sun[260], Zhe Yuan[260] & Jiajie Zhou[260]

[9]RWTH Aachen University, Aachen, Germany. [10]Aalto University, Espoo, Finland. [11]University of Helsinki, Helsinki, Finland. [12]Anhui University of Technology, Maanshan, AnHui, China. [13]Aix-Marseille University, Marseille, France. [14]AST Worldshaper, Hangzhou, China. [15]National Technical University of Athens, Athens, Attiki, Greece. [16]Athens University of Economics and Business, Athens, Attiki, Greece. [17]Liberal Arts and Science Academy High School, Austin, TX, USA. [18]The University of Texas at Austin, Austin, TX, USA. [19]Baltimore Underground Science Space, Baltimore, MD, USA. [20]Beijing City University, Beijing, China. [21]Beijing Foreign Studies University, Beijing, China. [22]BGI College, Shenzhen, China. [23]Ben-Gurion University of the Negev, Beer-Shiva, Israel. [24]Universitat Bielefeld, Bielefeld, Germany. [25]Bilkent University, Ankara, Turkey. [26]Institut Quimic de Sarria, Barcelona, Spain. [27]CHA University, Seongnam, South Korea. [28]Beijing Institute of Technology, Beijing, China. [29]Beijing Jianhua Experimental School, Beijing, China. [30]Beijing National Day School, Beijing, China. [31]Beijing Normal University, Beijing, China. [32]BOKU Vienna, Vienna, Austria. [33]Boston University, Boston, MA, USA. [34]University of British Columbia, Vancouver, BC, Canada. [35]University of Calgary, Calgary, Alberta, Canada. [36]Cardiff University, Cardiff, UK. [37]National Chung Cheng University, Min-Hsiung Chia-Yi, Minhsiung, Chiayi, Taiwan. [38]Chengdu Shishi High School, Chengdu, China. [39]Huayang High School, Chengdu, China. [40]Chalmers University of Technology, Gothenburg, Sweden. [41]China International Education Institute, Beijing, China. [42]Carnegie Mellon University in Qatar, Doha, Qatar. [43]Colorado School of Mines, Golden, CO, USA. [44]Columbia University, New York, NY, USA. [45]Cornell University, Ithaca, NY, USA. [46]China Pharmaceutical University, Nanjing, China. [47]Central South University, Changsha, China. [48]Colorado State University, Fort Collins, CO, USA. [49]Dalian University of Technology, Dalian, China. [50]Del Norte High School, San Diego, CA, USA. [51]Heinrich Heine University, Duesseldorf, Germany. [52]Universidad de las Fuerzas Armadas, Sangolqui, Ecuador. [53]East China University of Science and Technology, Shanghai, China. [54]University of Edinburgh, Edinburgh, UK. [55]Emory University Atlanta, Atlanta, GA, USA. [56]Ecole Polytechnique Federale de Lausanne, Lausanne, Switzerland. [57]ETH Zurich, Zurich and Basel, Basel, Switzerland. [58]University Paris-Saclay, Saint-Aubin, France. [59]University of Exeter, Exeter, UK. [60]Friedrich-Alexander-Universitat Erlangen-Nurnberg, Erlangen, Germany. [61]Fujian Normal University, Fuzhou, China. [62]Fudan University, Shanghai, China. [63]Guangdong Experimental High School, Guangzhou, Guangdong, China. [64]Georgia State University, Atlanta, GA, USA. [65]Gifu University, Gifu, Japan. [66]Shenzhen College Of International Education, Shenzhen, China. [67]University of Groningen, Groningen, Netherlands. [68]Guangdong Experimental High School, Guangzhou, China. [69]Hankuk Academy of Foreign Studies, Yongin, Korea. [70]Univesity of Hamburg, Hamburg, Germany. [71]Hubei University of Technology, Wuhan, China. [72]Hebrew University in Jerusalem, Jerusalem, Israel. [73]Hangzhou Foreign Language School, Hangzhou, China. [74]Zhejiang High Schools United, Hangzhou, China. [75]Tsuen Wan Public Ho Chuen Yiu Memorial College, Hong Kong, Hong Kong. [76]Po Leung Kuk Laws Foundation College, Hong Kong, Hong Kong. [77]St Stephen's College, Hong Kong, Hong Kong. [78]The University of Hong Kong, Hong Kong, Hong Kong. [79]Hong Kong University of Science and Technology, Hong Kong, Hong Kong. [80]United Christian College (Kowloon East), Hong Kong, Hong Kong. [81]Yan Oi Tong Tin Ka Ping Secondary School, Hong Kong, Hong Kong. [82]The Chinese University of Hong Kong, Hong Kong, Hong Kong. [83]Hubei University, Wuhan, China. [84]Huazhong University of Science and Technology, Wuhan, China. [85]Huazhong Agricultural University, Wuhan, China. [86]Institute of Chemical Technology, Mumbai, India. [87]Indian Institute of Science, Bangalore, India. [88]Indian Institute of Science Education and Research, Bhopal, India. [89]Indian Institute of Science Education and Research, Kolkata, India. [90]Indian Institute of Science Education and Research, Mohali, India. [91]IIT Delhi, New Delhi, India. [92]Indian Institute of Technology Kanpur, Kanpur, India. [93]Indian Institute of Technology Madras, Chennai, India. [94]Jiangnan University, Wuxi, China. [95]Jilin University, Changchun, China. [96]Julius-Maximilians Universitat,

Wurzburg, Germany. [97]Kanagawa Institute of Technology, Kanagawa, Japan. [98]King's College London, London, UK. [99]Korea University, Seoul, South Korea. [100]Lambert High School, Suwanee, GA, USA. [101]Leiden University, Leiden, The Netherlands. [102]University of Lethbridge, Lethbridge, Alberta, Canada. [103]Chinook High School, Lethbridge, Alberta, Canada. [104]Winston Churchill High School, Lethbridge, Alberta, Canada. [105]Catholic Central High School, Lethbridge, Alberta, Canada. [106]Lethbridge Collegiate Institute, Lethbridge, Alberta, Canada. [107]Texas Tech University, Lubbock, TX, USA. [108]Lund Tekniska Hogskola, Lund, Sweden. [109]Lund University, Lund, Sweden. [110]Macquarie University, Sydney, Australia. [111]Complutense University, Madrid, Spain. [112]Carlos III University, Madrid, Spain. [113]University of Manchester, Manchester, UK. [114]Philipps-University Marburg, Marburg, Germany. [115]McGill University, Montreal, Quebec, Canada. [116]McMaster University, Hamilton, ON, Canada. [117]METU Developmental Foundation High School, Ankara, Turkey. [118]University of Michigan, Ann Arbor, MI, USA. [119]Michigan State University, East Lansing, MI, USA. [120]Mingdao High School, Taichung City, Taiwan. [121]University of Minnesota, St. Paul, MN, USA. [122]University of Montpellier, Montpellier, France. [123]TU Munich, Garching, Germany. [124]LMU Munich, Munich, Germany. [125]Nanjing University, Nanjing, China. [126]Nanjing Agricultural University, Nanjing, China. [127]University Graz, Graz, Austria. [128]Graz University of Technology, Graz, Austria. [129]NAWI Graz, Graz, Austria. [130]National Chung Hsing University, Taichung, Taiwan. [131]National Chiao Tung University, Hsinchu, Taiwan. [132]Northeastern University, Shenyang, China. [133]Newcastle University, Newcastle, UK. [134]Nankai University, Tianjin, China. [135]Northwestern University, Evanston, IL, USA. [136]The University of Nottingham, Nottingham, UK. [137]Northwestern Polytechnical University, Xi'an, China. [138]National Tsing Hua University, HsinChu, Taiwan. [139]Norwegian University of Science and Technology, Trondheim, Norway. [140]Nanyang Technological University, Singapore, Singapore. [141]Nazarbayev University, Astana, Kazakhstan. [142]National University of Defense Technology, Changsha, China. [143]National University of, Singapore, Singapore. [144]Northwest University, Xi'an, China. [145]National Yang Ming University, Taipei, Taiwan. [146]New York University Abu Dhabi, Abu Dhabi, UAE. [147]Ocean University of China, Qingdao, China. [148]University of Oxford, Oxford, UK. [149]CRI Paris, Paris, France. [150]Institut Pasteur, Paris, France. [151]Peking University, Beijing, China. [152]University of Pittsburgh, Pittsburgh, PA, USA. [153]Purdue University, West Lafayette, IN, USA. [154]Queen's University, Kingston, ON, Canada. [155]RDFZ, Beijing, China. [156]Rajalakshmi Engineering College, Chennai, India. [157]Bielefeld University, Bielefeld, Germany. [158]Einstein Gymnasium, Rheda-Wiedenbruck, Germany. [159]Rose-Hulman Institute of Technology, Terra Haute, IN, USA. [160]Rice University, Houston, TX, USA. [161]Ramnarain Ruia Autonomous College, Mumbai, India. [162]Stony Brook School, Stony Brook, NY, USA. [163]He County First Middle School, Shanghai, China. [164]South China Agricultural University, Guangzhou, China. [165]Sichuan University, Chengdu, China. [166]South China University of Technology, Guangzhou, China. [167]Shandong University, Qingdao, China. [168]Shenzhen Foreign Languages School, Shenzhen, China. [169]ShanghaiTech University, Shanghai, China. [170]SHSBNU, Beijing, China. [171]Shanghai High School International Division, Shanghai, China. [172]Shanghai Foreign Language School, Shanghai, China. [173]Shanghai Pinghe School, Shanghai, China. [174]Shanghai Qibao Dwight High School, Shanghai, China. [175]Shenzhen College of International Education, Shenzhen, China. [176]Shanghai Jiao Tong University, Shanghai, China. [177]State Key Laboratory of Microbiological Technology, Shan Dong University, Qingdao, China. [178]Second Military Medical University, Shanghai, China. [179]Shenzhen Middle School, Shenzhen, China. [180]Sorbonne Universite, Paris, France. [181]Shenzhen Senior High School, Shenzhen, China. [182]Shenzhen Institute of Technology, Shenzhen, China. [183]University of St Andrews, St, Andrews, UK. [184]Stanford University, Stanford, CA, USA. [185]Brown University, Providence, RI, USA. [186]Rhode Island School of Design, Providence, RI, USA. [187]KTH Royal institute of Technology, Stockholm, Sweden. [188]Karolinska Institutet, Solna, Stockholm, Sweden. [189]Konstfack, Stockholm, Sweden. [190]Stony Brook University, Stony Brook, NY, USA. [191]University of Stuttgart, Stuttgart, Germany. [192]Shanghai United International School, Shanghai, China. [193]Sun Yat-sen University, Guangzhou, China. [194]Shenzhen University, Shenzhen, China. [195]Readiness Acceleration and Innovation Network, Tacoma, WA, USA. [196]University of Tartu, Tartu, Estonia. [197]Taipei American School, Taipei, Taiwan. [198]Instituto Tecnologico y de Estudios Superiores de Monterrey - Campus Chihuahua, Chihuahua, Mexico. [199]Tecnologico de Monterrey Campus Monterrey, Monterrey, Mexico. [200]Tecnologico de Monterrey CEM, Atizapan de Zaragoza, Ciudad Lopez Mateos, Mexico. [201]Tecnologico de Monterrey Campus Guadalajara, Guadalajara, Mexico. [202]Aristotle University of Thessaloniki, Thessaloniki, Greece. [203]Tongji University, Shanghai, China. [204]University of Toronto, Toronto, ON, Canada. [205]Institut National des Sciences Appliquees de Toulouse, Toulouse, France. [206]Universite Toulouse III Paul Sabatier de Toulouse, Toulouse, France. [207]Technische Universitaet Darmstadt, Darmstadt, Germany. [208]Eindhoven University of Technology, Eindhoven, The Netherlands. [209]Delft University of Technology, Delft, The Netherlands. [210]Tianjin Univeresity of Science and Technology, Tianjin, China. [211]University of Alberta, Edmonton, Alberta, Canada. [212]University of California Davis, Davis, CA, USA. [213]University of California San Diego, La Jolla, USA. [214]University of Chinese Academy of Sciences, Beijing, China. [215]University of Chicago, Chicago, IL, USA. [216]University of Chile, Santiago, Chile. [217]University College London, London, UK. [218]Universite Catholique de Louvain, Louvian-la-Neuve, Belgium. [219]University of Connecticut, Storrs, CT, USA. [220]University of Copenhagen, Copenhagen, Denmark. [221]University of Electronic Science and Technology of China, Chengdu, China. [222]University of Florida, Gainesville, FL, USA. [223]University of Georgia, Athens, GA, USA. [224]Universitas Indonesia, Depok, West Java, Indonesia. [225]University of Oslo, Oslo, Norway. [226]University of Iowa, Iowa City, IA, USA. [227]University of Illinois, Champaign-Urbana, IL, USA. [228]Universite Laval, Quebec, QC, Canada. [229]University of La Verne, La Verne, CA, USA. [230]University of Maryland, College Park, MD, USA. [231]University of Nebraska - Lincoln, Lincoln, NE, USA. [232]Universidade Estadual Paulista, Araraquara, Sao Paolo, Brazil. [233]University of New South Wales, Randwick, Australia. [234]Pompeu Fabra University, Barcelona, Spain. [235]Center for Genomic Regulation, Barcelona, Spain. [236]Uppsala University, Uppsala, Sweden. [237]Carroll High School, Dayton, OH, USA. [238]United States Military Academy, West Point, NY, USA. [239]Universidade de Sao Paulo, Sao Paulo, Brazil. [240]University of Science and Technology, Beijing, China. [241]University of Science and Technology of China, Hefei, China. [242]University of Tennessee, Knoxville, TN, USA. [243]Utrecht University, Utrecht, The Netherlands. [244]Universitat Politecnica de Valencia, Valencia, Spain. [245]Vilnius University, Vilnius, Lithuania. [246]University of Virginia, Charlottesville, VA, USA. [247]University of Warwick, Coventry, UK. [248]University of Washington, Seattle, WA, USA. [249]Washington University in St. Louis, St. Louis, MO, USA. [250]University of Waterloo, Waterloo, ON, Canada. [251]University of Westminster, London, UK. [252]Wuhan University, Wuhan, China. [253]College of William and Mary, Williamsburg, VA, USA. [254]Worldshaper Wuhan, Hangzhou, China. [255]Xi'an Jiaotong-Liverpool University, Suzhou, China. [256]Xi'an Jiaotong University, Xi'an, China. [257]Xiamen University, Xiamen, China. [258]Yale University, New Haven, CT, USA. [259]Zhejiang University, Hangzhou, China. [260]Zhejiang University of Technology, Hangzhou, China.

