## [Peer Review File · Communications Biology]

Reviewers' comments:

Reviewer #1 (Remarks to the Author):

Optical density (OD) method measures turbidity associated directly with bacterial growth which is rapid, low cost and nondestructive, however the measurement is based on the amount of light scattered by the culture rather than the amount of light absorbed. Spectrophotometers are not optimized for light scattering measurements resulting in differences in measured absorbance in different instruments. In this study, the authors compared three OD calibration protocols with an interlaboratory study by using various strains of *E. coli* engineered to express varying levels of GFP. They demonstrated that serial dilution of silica microspheres method is the best of these tested protocols, allowing highly precise accurate, and robust calibration that is easily assessed for quality control and can also evaluate the effective linear range of an instrument. The overall idea of the research is attractive. The experiments were well designed and executed. The manuscript was well prepared.

Reviewer #2 (Remarks to the Author):

I would like to start this review by stating that I am very much in favor of initiatives that help standardizing methods in (micro)biology. In addition, I am impressed by the scale of this initiative (244 teams!)

However, at the same time I must admit that after reading this manuscript, I don't understand what the actual goal of this study is and what microbiologists around the world would practically do with it. After all, microbiologists know that correlating OD measurements with cell numbers is difficult, which is why most of use microbiologists would set up OD/CFU calibration curves for every organisms we work with. We are aware that this information is not very 'portable' but as every lab can easily determine which OD corresponds to a certain number of CFU in their specific lab environment (with their specific spectrophotometer etc) this doesn't seem to be a big deal.

The issue may be with the way this manuscript is written – again, after reading it I have no idea what I would need to change to do better and maybe the authors should consider rewriting the manuscript with the end user in mind.

Reviewer #3 (Remarks to the Author):

In the Methods manuscript "Robust Estimation of Bacterial Cell Count from Optical Density", Beal and colleagues compared the accuracy and performance of three optical density calibration protocols to estimate bacterial counts. These protocols are: (1) a colony-forming unit assay, (2) colloidal silica and water; and (3) silica microspheres. The authors expanded their analysis using tools and approaches developed previously (PLoS ONE 13(6): e0199432; PLoS ONE 11(3): e0150182). Based on data collected from 244 teams of high school, undergraduate and graduate students, the authors found that silica microspheres provided the most precise calibration. This calibration protocol was as effective as using fluorescein previously optimized by the authors. The colony-forming unit assay, as expect, is the least reliable and accurate for calibration.

This manuscript is technically sound and serves to remind scientists who are using Optical Density as a measure of bacterial numbers that certain calibration tools are better than others. The manuscript is also useful to high school and university students who are beginning to appreciate the strengths and

weaknesses of different calibration methods to estimate bacterial numbers. I only have a few minor comments which should be clarified prior to publication of this manuscript:

1. In the "Experimental Data Collection" section where the colloidal silica LUDOX CL-X and silica microspheres are described, it would be helpful to highlight the differences between colloidal silica and silica microspheres in terms of particle size or diameter, and optical properties.
2. What are the main differences in the eight constructs? Exactly why do you need all eight in this study? The current information "to give a range of GFP expression" doesn't give a strong rationale of why you need them for this study (Page 3). Although some of this information may have been outlined in the supplementary data or in their previous publications, the basic details of why you need the eight constructs for this study should be clearly stated in the "Experimental Data Collection" section.
3. Why were 0 h and 6 h the only two time points chosen? This selection seems arbitrary. The robustness of the study would have been enhanced if a range of time points were chosen.
4. Were teams asked to do independent repeats in addition to technical repeats in their sampling?
5. Were data from all 244 teams used and plotted in the figures? In the relevant figure legends, please state the number of teams in which the data were derived.
6. Figure 1: the labelling of the expression constructs in Panel A and dilution factors in other panels are poor and not visible. Please use a legible font size in Figure 1.
7. Figure 5. The legends in the panel should not overlap the graph.
8. Materials and Methods: please add catalogue numbers for LUDOX, fluorescein and silica beads alongside their respective companies.

Response to Reviewers

Reviewer #1

Optical density (OD) method measures turbidity associated directly with bacterial growth which is rapid, low cost and nondestructive, however the measurement is based on the amount of light scattered by the culture rather than the amount of light absorbed. Spectrophotometers are not optimized for light scattering measurements resulting in differences in measured absorbance in different instruments. In this study, the authors compared three OD calibration protocols with an interlaboratory study by using various strains of E. coli engineered to express varying levels of GFP. They demonstrated that serial dilution of silica microspheres method is the best of these tested protocols, allowing highly precise accurate, and robust calibration that is easily assessed for quality control and can also evaluate the effective linear range of an instrument. The overall idea of the research is attractive. The experiments were well designed and executed. The manuscript was well prepared.

Thank you!

Reviewer #2:

I would like to start this review by stating that I am very much in favor of initiatives that help standardizing methods in (micro)biology. In addition, I am impressed by the scale of this initiative (244 teams!)

Thank you!

However, at the same time I must admit that after reading this manuscript, I don't understand what the actual goal of this study is and what microbiologists around the world would practically do with it. After all, microbiologists know that correlating OD measurements with cell numbers is difficult, which is why most of use microbiologists would set up OD/CFU calibration curves for every organisms we work with. We are aware that this information is not very 'portable' but as every lab can easily determine which OD corresponds to a certain number of CFU in their specific lab environment (with their specific spectrophotometer etc) this doesn't seem to be a big deal. The issue may be with the way this manuscript is written – again, after reading it I have no idea what I would need to change to do better and maybe the authors should consider rewriting the manuscript with the end user in mind.

One of the key results in this paper is that OD/CFU relationships are not very reliable or replicable, while OD/particle relationships following the introduced protocol can be much more precise. We are thus recommending use of silica microsphere calibration within the linear range of OD measurements, which should enable more effective use of OD data for estimation of actual cell count, comparison of plate reader measurements with single-cell measurements such as flow cytometry, improved replicability, and better cross-laboratory comparison of data. We have added clarifying text emphasizing this recommendation to the abstract, introduction and discussion sections.

Reviewer #3

In the Methods manuscript “Robust Estimation of Bacterial Cell Count from Optical Density”, Beal and colleagues compared the accuracy and performance of three optical density calibration protocols to estimate bacterial counts. These protocols are: (1) a colony-forming unit assay, (2) colloidal silica and water; and (3) silica microspheres. The authors expanded their analysis using tools and approaches developed previously (PLoS ONE 13(6): e0199432; PLoS ONE 11(3): e0150182). Based on data collected from 244 teams of high school, undergraduate and graduate students, the authors found that silica microspheres provided the most precise calibration. This calibration protocol was as effective as using fluorescein previously optimized by the authors. The colony-forming unit assay, as expected, is the least reliable and accurate for calibration.

This manuscript is technically sound and serves to remind scientists who are using Optical Density as a measure of bacterial numbers that certain calibration tools are better than others. The manuscript is also useful to high school and university students who are beginning to appreciate the strengths and weaknesses of different calibration methods to estimate bacterial numbers.

Thank you! Unfortunately, our observations are that it is not just high school and university students, but the vast majority of professional researchers as well, give that most publications involving OD measurements still do not attempt to relate these measurements to cell counts.

I only have a few minor comments which should be clarified prior to publication of this manuscript:

1. In the “Experimental Data Collection” section where the colloidal silica LUDOX CL-X and silica microspheres are described, it would be helpful to highlight the differences between colloidal silica and silica microspheres in terms of particle size or diameter, and optical properties.

We have added in this information, where LUDOX particles are <50nm vs. the 961um diameter of the microspheres. Given that LUDOX is already disqualified from direct comparison by vast difference between the size of its particles and the size of cells, we do not detail the optical properties of LUDOX. We have added the refractive index of the silica microspheres and its comparison with typical E. coli values.

2. What are the main differences in the eight constructs? Exactly why do you need all eight in this study? The current information “to give a range of GFP expression” doesn’t give a strong rationale of why you need them for this study (Page 3). Although some of this information may have been outlined in the supplementary data or in their previous publications, the basic details of why you need the eight constructs for this study should be clearly stated in the “Experimental Data Collection” section.

Each was chosen for a particular reason, as follow: the positive and negative controls and the J23101, J23106, and J23117 promoters were chosen based on their prior successful use in the 2016 iGEM interlaboratory study as controls and “high”, “medium”, and “low” test levels respectively. Beyond these, J23100 and J23104 were chosen as potential alternatives for J23101

(about which there were previous anecdotal reports of difficulty), and J23116 was chosen as an intermediate value in the large gap in expression levels between J23106 and J23117. We have added this additional explanation at the point where the constructs are introduced.

3. Why were 0 h and 6 h the only two time points chosen? This selection seems arbitrary. The robustness of the study would have been enhanced if a range of time points were chosen.

In fact, we are primarily using only one time point, as time series is not the focus of this study: 6 hours was chosen as a period sufficient for significant exponential growth, and 0 hours used only for comparison to exclude samples that failed to grow well. We have included this clarification at the point where the time points are introduced.

4. Were teams asked to do independent repeats in addition to technical repeats in their sampling?

Each team was asked to do two biological replicates for each construct, per the description in the Experimental Data Collection section and the supplementary protocol, meaning there were a total of 8 replicates: 2 biological x 4 technical. We have added clarifying text emphasizing this.

5. Were data from all 244 teams used and plotted in the figures? In the relevant figure legends, please state the number of teams in which the data were derived.

Figure 2 and 3 use data from all 244 teams, while Figure 4 and 5 use only the valid data as specified in “Criteria for Valid E. coli Data” in the methods. As the exclusions affect each calibration/measurement condition and each construct differently, we provide the number of teams datasets used for each combination in a new supplementary data.

6. Figure 1: the labelling of the expression constructs in Panel A and dilution factors in other panels are poor and not visible. Please use a legible font size in Figure 1.

We have adjusted the figure and increased font size for increased readability.

7. Figure 5. The legends in the panel should not overlap the graph.

We believe the current position would be acceptable because it does not overlap any of the actual data points, but have in any case moved the legend outside of the plot.

8. Materials and Methods: please add catalogue numbers for LUDOX, fluorescein and silica beads alongside their respective companies.

We have added the catalog numbers, as requested.

Reviewer #4

I would not recommend publishing this manuscript in its current form, and not with its current title for the following main reason (see also my comment to line 21-23). Calibration of optical density measurements to cell concentration (or biomass) is influenced by several factors, such as cell size. To have a general title like the authors propose one would need to repeat what they do

for cells of all kinds of different sizes, for conditions that might change the indexes of refraction (of the media and the cells too). The cells the authors are using most likely fall under a narrow range of sizes and the indexes of refraction stay fixed. I think the protocols they propose and the comparisons between different methods are valuable but it needs to be clearly put into context. The title also should not be kept as it is, it suggests a general result, where this is not the case. If kept, someone might apply the same microsphere calibration protocol for cells whose size will not render spheres to be the best approximation for a cell. I, therefore, suggest clarifying this in the text and changing the title accordingly (bacterial cell should be E. coli cells grown to exp phase in media with doubling time X, or the title should mention that the protocol is for specific sizes of cells and these numbers given in the main text). The introduction to the problem should be accordingly modified.

Although we have already acknowledged in the text that the microsphere protocol is only expected to be applicable to cells that can be reasonably approximated by microspheres and within the linear range of OD (which includes the effects of media, per Stevenson et al.), we agree with the reviewer that these limitations could be more clearly emphasized. We have accordingly added clarifying text in the introduction and discussion. Likewise, we have added information about the refractive index of the microspheres and *E. coli* that indicates these are a reasonable approximation across a range of typical growth conditions, as well as the fact that many other common bacteria have quite similar refractive index values. While there are, indeed, many things that can affect refraction index, Stevenson et al. has already shown that the magnitude of these effects relative to the precision of the protocol is relatively small and established the scalability of the relevant physical phenomena.

Below are some specific comments:

suggest removing 'absorbance' in line 6

Agreed that this word was redundant; removed.

21-23 -comment to this sentence, this is because it is impossible to do given the nature of the measurement, the reason why you were able to calibrate well here is that you used very similar strains and a very narrow range of conditions, so this statement is misleading. As is the title, I can not recommend publication with such title, it needs to be more specific, i.e. it needs to indicate which bacteria you used (index of refraction), the range of their dimensions and the index of refraction of your media

See above response to reviewer's first comment.

-silica or plastic? (what is the index of refraction of these speheres? apart from the size, this is important)

As stated in the text, they are silica microspheres, not plastic. As noted above, we have also added refractive index information.

line 72 -'be' instead of 'by'

Thank you; fixed this typo.

-line 84- non-viable cells can still scatter, so I would say this is why this protocol is not the best

Agreed, as we note with our statement about the lack of a conversion from CFU to number of cells.

-line 97, not clear why not, the colloid solution usually come with a know concentration (within the manufacturers' error of estimate)

Our phrasing was unclear: the intention was to explain that we cannot estimate cell count from LUDOX because the particles are far smaller than cells (<50nm). We have clarified the statement.

-line 102- 'similar optical properties to E.coli' is not accurate enough, again E.coli can change size drastically, here what needs to be said explicitly is the size, index of refraction of E.coli and the media in which this is applicable

The phrasing of the paper needs to be changed too, as it is not it is misleading

As noted above, we have added refractive index information and made the range of expected applicability more explicit.

-337 should this not be plotted in SI somewhere so that the linear range of the APD the instrument is using is clear, I would suggest including the rest of the instrument calibration in that figure too (gating etc) and since there already is an SI section on flow cytometry protocols

Since the flow cytometry calibration applied here is long-established standard practice, we do not believe adding all 68 calibration charts for 17 different flow cytometers would be particularly useful. To assist readers who may be unfamiliar with these practices, however, we have added additional information about the processing, a reference to the relevant technical specification and reference NIST study, as well as supplementing the prototypical example of gating with prototypical examples of autofluorescence and bead calibration.

350-351 how is 'too numerous to count' defined? at least two colonies that are touching or?

Participants were instructed they could report anything over 300 colonies to be too numerous to count; we have added a note to that effect in the methods section.

363-364, 4 would be better, how would results change if you took 4 levels instead?

Supplementary Figure 1 shows that the number datasets that would be affected is quite small (3.3% or 0.4%). This has, unsurprisingly, little effect on the results. We have added a clarifying note that emphasizes the information in Supplementary Figure 1.

382 sigma not defined

We used the standard μ/σ convention for mean and standard deviation; to avoid confusion, we have now added an explicit definition statement as well.

219- I agree that J23104 is different, not sure I agree that this is the case with J23101, why are the authors saying this is significantly more variable. But this difference in strain to strain variability points again to my main criticism of this paper

This comment appears to refer to the levels of variability in Figure 4, while the text is actually referring to the variability in reliable colony growth, as indicated by the numbers of failed samples in Supplementary Figure 2. We have added clarifying text indicating that the variability in question is the fact that many teams' samples failing to grow for these constructs, while other constructs grew much more reliably.

231- as far as I am understanding the count from flow cytometry is used as the ground truth of the cell number. I find this somewhat concerning, especially since full calibration of the instrument is not given in SI. Flow cytometry is also a scattering measurement unless the authors are using the devices with integrating imaging which is not the cases. That means that the numbers are determined based on scattering by spheres, and that will be affected by many of the same things OD's are affected by. So here those measurements should be compared with a form of direct counting in the microscope.

In this manuscript, flow cytometry is used to measure per-cell fluorescence, not cell count, and thus does not suffer from any of these limitations. The accuracy of cell count estimated from OD may then be checked by comparing the per-cell fluorescence estimated by dividing bulk fluorescence by estimated cell count to the per-cell fluorescence measured directly via flow cytometry. We have added additional clarifying text to help avoid confusion.

241- not sure this is the case, if they contribute to scattering in the spectrophotometer, they will for the scattering of the laser too and how well this can be gated out depends on the size of Debris, this requires some plots I think. But I agree in exp phase there should not be much Debris, however, the distribution of filamentous cells between different strains could be different. Again in exp phase, there should not be many of those but some sort of visual inspection between the strains would be useful.

Again, scattering due to particle size/shape is not a concern when using flow cytometry to measure per-cell fluorescence. For gating, however, we have weakened our statement to note that typically only the vast majority of debris particles that are debris particles that can be excluded. This minor adjustment, however, does not make any significant alteration in the overall statement regarding two anticipated distortions of opposing sign.